# Cancer-associated cells release citrate to support tumour metastatic progression

Konstantin Drexler[1],*, Katharina M Schmidt[2],*, Katrin Jordan[2], Marianne Federlin[3], Vladimir M Milenkovic[4], Gerhard Liebisch[5], Anna Artati[6], Christian Schmidl[7], Gregor Madej[8], Janina Tokarz[6], Alexander Cecil[6], Wolfgang Jagla[9], Silke Haerteis[10], Thiha Aung[10,11], Christine Wagner[2], Maria Kolodziejczyk[2], Stefanie Heinke[2], Evan H Stanton[2], Barbara Schwertner[1], Dania Riegel[7], Christian H Wetzel[4], Wolfgang Buchalla[3], Martin Proescholdt[12], Christoph A Klein[13], Mark Berneburg[1], Hans J Schlitt[2], Thomas Brabletz[14], Christine Ziegler[8], Eric K Parkinson[15], Andreas Gaumann[9], Edward K Geissler[2], Jerzy Adamski[6,16,17], Sebastian Haferkamp[1],*, Maria E Mycielska[2],*

Citrate is important for lipid synthesis and epigenetic regulation in addition to ATP production. We have previously reported that cancer cells import extracellular citrate via the pmCiC transporter to support their metabolism. Here, we show for the first time that citrate is supplied to cancer by cancer-associated stroma (CAS) and also that citrate synthesis and release is one of the latter's major metabolic tasks. Citrate release from CAS is controlled by cancer cells through cross-cellular communication. The availability of citrate from CAS regulated the cytokine profile, metabolism and features of cellular invasion. Moreover, citrate released by CAS is involved in inducing cancer progression especially enhancing invasiveness and organ colonisation. In line with the in vitro observations, we show that depriving cancer cells of citrate using gluconate, a specific inhibitor of pmCiC, significantly reduced the growth and metastatic spread of human pancreatic cancer cells in vivo and muted stromal activation and angiogenesis. We conclude that citrate is supplied to tumour cells by CAS and citrate uptake plays a significant role in cancer metastatic progression.

## Introduction

Citrate is a central metabolite used by cancer cells for fatty acid synthesis (Wang et al, 2016). One path of citrate synthesis is via the reverse Krebs cycle (Metallo et al, 2011). We have recently discovered that cancer cells can also import extracellular citrate by using a plasma membrane citrate transporter (pmCiC; Mycielska et al, 2018). This transporter has been cloned from the citrate releasing prostate secretory epithelial cells (Mazurek et al, 2010) and determined to change citrate transport direction, depending on the cell type in which it is expressed (Mazurek et al, 2010; Mycielska et al, 2018). Moreover, extracellular citrate can directly promote cancer proliferation (Petillo et al, 2020) and alter metabolism in vitro (Mycielska et al, 2018). In addition, pmCiC was found to be overexpressed in human cancer cells compared with their benign counterparts in vivo, where it was found to correlate with tumour grade, and was particularly increased at the invasion front and metastatic sites of tumours of different origin (Mycielska et al, 2018). Daily application of a specific pmCiC inhibitor, gluconate, slowed xenograft growth in vivo by incompletely understood mechanisms (Mycielska et al, 2018, 2019).

There is now extensive evidence that human cancer cells can induce neighbouring stromal cells to produce proteins and metabolites that aid tumour development, progression, and drug resistance (Nazemi & Rainero, 2020). The metabolites documented include nucleotides, lactate, and amino acids such as serine and glutamine (Nazemi & Rainero, 2020), but the role of citrate in crosstalk between tumour cells and the tumour microenvironment has hitherto not been investigated in this regard. One route of citrate supplied to cancer could be through blood. Citrate blood concentration is around 200 $\mu M$ (Mycielska et al, 2015). However, blood supply to growing tumours is

---

[1]Department of Dermatology, University Medical Centre, Regensburg, Germany   [2]Department of Surgery, University Medical Center, Regensburg, Germany   [3]Department of Conservative Dentistry and Periodontology, University Medical Center, Regensburg, Germany   [4]Department of Psychiatry and Psychotherapy, University of Regensburg, Regensburg, Germany   [5]Institute of Clinical Chemistry and Laboratory Medicine, Regensburg University Hospital, Regensburg, Germany   [6]Research Unit Molecular Endocrinology and Metabolism, Helmholtz Zentrum München, German Research Centre for Environmental Health, Neuherberg, Germany   [7]Regensburg Center for Interventional Immunology, Regensburg, Germany   [8]Department of Structural Biology, Institute of Biophysics and Physical Biochemistry, University of Regensburg, Regensburg, Germany   [9]Institute of Pathology, Kaufbeuren-Ravensburg, Kaufbeuren, Germany   [10]Institute for Molecular and Cellular Anatomy, University of Regensburg, Regensburg, Germany   [11]Center of Plastic, Aesthetic, Hand and Reconstructive Surgery, University of Regensburg, Regensburg, Germany   [12]Department of Neurosurgery, University Hospital Regensburg, Regensburg, Germany   [13]Experimental Medicine and Therapy Research, University of Regensburg, Regensburg, Germany   [14]Department of Experimental Medicine 1, Friedrich-Alexander-University Erlangen, Erlangen, Germany   [15]Centre for Immunobiology and Regenerative Medicine, Blizard Institute, Barts and The London School of Medicine and Dentistry, London, UK   [16]Lehrstuhl für Experimentelle Genetik, Technische Universität München, Munich, Germany   [17]Department of Biochemistry, Yong Loo Lin School of Medicine, National University of Singapore, Singapore

Correspondence: sebastian.haferkamp@ukr.de; maria.mycielska@ukr.de
*Konstantin Drexler, Katharina M Schmidt, Sebastian Haferkamp, and Maria E Mycielska contributed equally to this work

---

   

often restricted and metastatic growth requires new blood vessel formation. Another source of extracellular citrate could be tumour-surrounding tissue known to have elevated citrate levels, for example, benign prostate epithelium (Eidelman et al, 2017) or astrocytes in the brain (reviewed by Mycielska et al [2015]).

Cancer-associated fibroblasts (CAFs) have already been shown to support cancer metabolism by releasing several growth factors, enzymes, and metabolites (Sakamoto et al, 2019). The main objective of this study was to determine whether CAFs/cancer-associated stromal cells are the source of extracellular citrate to cancer cells and the effects of extracellular citrate on cancer metastatic progression.

We report evidence that CAFs are a substantial source of citrate for cancer cells and that CAFs release citrate via the pmCiC. Importantly, the metabolic activity of CAFs depends strongly on the availability of citrate to cancer cells suggesting that citrate synthesis and release is one of the main metabolic tasks of CAFs. We also show that extracellular citrate plays a role in the induction of an invasive phenotype and subsequent organ colonisation by cancer cells. Consistent with this observation, daily treatment of mice with gluconate (a specific inhibitor of pmCiC in cancer cells) significantly reduced cancer spread, stromal transformation, angiogenesis, and increased immune infiltration. Citrate is, therefore, a critical element of the cross-talk between cancer cells and cancer-associated surrounding tissue and the presence of extracellular citrate is crucial for metastatic progression.

## Results

### Availability of citrate to cancer cells determines metabolic activity of CAFs

Human fibroblasts grown in control media versus media conditioned by prostate cancer cell line PC-3M (CAFs) exhibited distinctly different metabolic characteristics (Figs S1, S2, and 3A). One of the metabolites which was released at a high level by CAFs was citrate. CAFs also released significant amounts of panthotenate, biotin, and proline. All these metabolites are involved in the support of fatty acid synthesis (panthotenate), their metabolism (biotin), or protein synthesis (proline). As observed earlier (Romero et al, 2015), we also found that CAFs released increased amounts of several amino acids. Analysis of conditioned media from cancer cells and unconditioned media showed that neither of them contained citrate (Figs S1 and S3A). Therefore, it can be deduced that citrate present in the media from transformed fibroblasts is exclusively due to the synthesis and release from CAFs. Furthermore, the levels of pyruvate which can be used by cells for intracellular citrate synthesis remained the same in the conditioned media from cancer cells and fibroblasts, suggesting that pyruvate does not play a significant role in CAF's support of cancer metabolism.

Release of citrate indicated that this metabolite might play an important role in the crosstalk between cancer cells and CAFs. Therefore, we decided to study the differences in the way stroma is transformed in response to cancer cells preincubated without or with extracellular citrate (CM PC-3M−cit and CM PC-3M+cit, respectively; Fig

S3B and C). Cancer cells were preincubated with, or without citrate for 48 h, and the medium was removed and replaced with the citrate-free media used for conditioning for another 48 h. To determine the effects of extracellular citrate only, under all experimental conditions for cancer cells and fibroblasts and at each step of the experimental procedure, glucose and glutamine were supplied in excess to allow for unlimited intracellular citrate synthesis (Fig S3). Using the Seahorse instrument, we determined that CAFs transformed with the CM PC-3M−cit media showed a significant increase in their $O_2$ consumption and glycolytic activity as compared with CAFs transformed with the CM PC-3M+cit media (Fig 1A). Increased $O_2$ consumption was insensitive to mitochondrial blockers, suggesting non-mitochondrial use of the oxygen. Correspondingly, there was also a significant decrease in mitochondrial ATP synthesis. CAFs transformed by CM PC-3M−cit expressed high level of glycolysis, whereas CAFs transformed with the CM PC-3M+cit media used predominantly OXPHOS to synthesise ATP. Media from fibroblasts grown in either CM PC-3M−cit or CM PC-3M+cit were then examined for citrate content. We found that the level of citrate in media from fibroblasts stimulated with CM PC-3M−cit were significantly higher (~300 $\mu$M, Fig 1B) than the concentration of citrate produced by fibroblasts stimulated with CM PC-3M+cit (~200 $\mu$M). These data stay in line with the metabolic changes (Fig 1A) where fibroblasts releasing more citrate showed higher metabolic activity and a shift towards glycolysis.

It is important to note that no citrate was present in either of the conditioned media used to transform fibroblasts (Fig S3); therefore, the obtained results are exclusively due to the presence/absence of extracellular citrate in the preincubation media of cancer cells.

Altogether, these data indicate that one of the crucial metabolites released by CAFs in support of cancer cells is citrate. The amount of citrate released and in consequence CAFs metabolic activity is determined by the availability of citrate to cancer cells.

Transformation of fibroblasts with conditioned media from the PC-3M or L3.6pl (human prostate or pancreatic cancer cells, respectively) resulted in the expression of pmCiC (Fig 1C; able to transport citrate to the extracellular space as shown already in the case of benign prostate epithelial cells; Mazurek et al, 2010). Similarly, hepatic stellate cells, known to support metastases to liver (Milette et al, 2017), incubated with conditioned media from human prostate cancer PC-3M or human pancreatic L3.6pl cells showed a significant expression of the pmCiC (Fig 1C). pmCiC expression was very low/negligible in fibroblasts and all hepatic stellate cell lines measured under control conditions (where no conditioned media from cancer cells were present; Fig 1C). It is important to note that although LX2 and HSCh[TERT] are permanently activated human hepatic stellate cell lines, they only express pmCiC after incubation with conditioned media from cancer cells. Therefore, it can be concluded that citrate synthesis and release from CAFs takes place only when it is induced by cancer cells.

To investigate the functional consequences of these metabolic alterations, we next determined the differential expression of cytokines under the above conditions. Cancer cells grown in the presence of extracellular citrate largely released cytokines known to stimulate tumour metastatic activity like MCP1 (Li et al, 2013), IL-8 (Yuan et al, 2005), and RANTES (Azenshtein et al, 2002; Figs 1D and S4). On the other hand, cancer cells deprived of citrate released cytokines largely associated with stromal transformation (Figs 1D

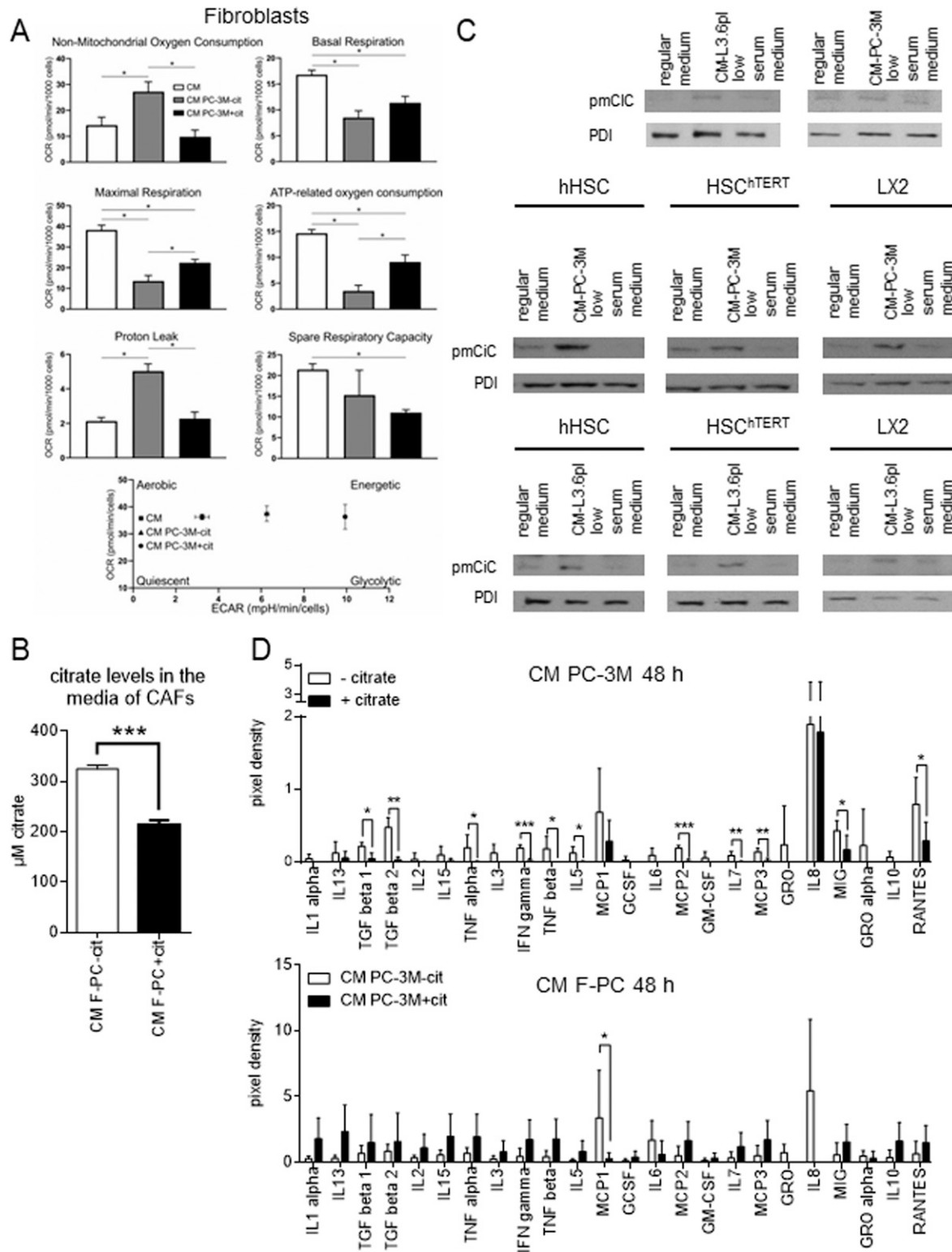

**Figure 1. Cancer-associated fibroblasts release citrate through pmCiC.**
**(A)** Seahorse analysis of the metabolic characteristics and the energy map of fibroblasts grown in media used for conditioning, CM PC-3M−cit or CM PC-3M+cit (n = 4). The graphs show differences between normal fibroblasts (white), fibroblasts transformed with CM PC-3M−cit (grey), and CM PC-3M+cit (black). **(B)** The graph depicts citrate levels in the media from fibroblasts stimulated with CM PC-3M−cit (white) and CM PC-3M+cit (black). **(C)** pmCiC expression in fibroblasts and different hepatic stellate cell lines grown in regular medium, low serum medium or transformed with conditioned media from PC-3M cells or L3.6pl cells. **(D)** The graphs show selected cytokines (n = 5) released from PC-3M cancer cells preincubated without or with extracellular citrate for 48 h (CM PC-3M−cit CM PC-3M+cit, respectively; upper panel) and human primary skin fibroblasts transformed by CM PC-3M−cit CM PC-3M+cit (CM F-PC−cit and CM F-PC+cit, respectively; lower panel).

and S4A–H). These included MCP2,3, IL-5, TGFβs, and GROs (although GROs are not statistically different between the groups). These cytokines were either absent in the media from cancer cells preincubated with citrate or expressed at a low level. GRO and MCP2 and 3 are known to transform stroma (Yang et al, 2006; Argyle & Kitamura, 2018; Sun et al, 2018), whereas IL-5 has been shown to modulate tumour environment (Zaynagetdinov et al, 2015). TGFβs act on the tumour microenvironment by stimulating the transformation of fibroblasts into myofibroblasts (Papageorgis & Stylianopoulos, 2015). Release of these cytokines suggests that cancer cells deprived of citrate concentrate their efforts on transforming the surrounding stroma to get metabolic support.

In response to CM PC-3M+cit, fibroblasts released several cytokines (Figs 1D and S4) known to support cancer cell's metastatic behaviour such as IL13 (known to support cancer cell survival and metastasis; Suzuki et al, 2015), MCP2,3 (Zhang et al, 2020), and RANTES (Singh et al, 2018). CM PC-3M–cit also stimulated release of different cytokines from fibroblasts but they tended (although not statistically different) to be at a lower level (Figs 1D and S4).

Next, we investigated whether citrate induced any changes in PC-3M metabolism that might account for the above phenotypes. PC-3M cells incubated with extracellular citrate showed changes at the metabolite levels. We have focused on the differences between the –citrate versus +citrate groups. In the presence of extracellular citrate, cancer cells decreased intracellular levels of metabolites associated with intracellular citrate synthesis (reviewed by Haferkamp et al [2020]). In particular, there was a comparatively low level of intracellular glutamate, glutamine, but also glycine and serine (Fig 2A). Consistently, cancer cells grown in the media not supplemented with extracellular citrate showed an increased release of citrate-synthesis related metabolites such as lactate, serine, or acetylcarnitine (Fig S1). Compared with cancer cells preincubated without citrate, extracellular citrate supply decreased overall intracellular levels of metabolites shown best by total amounts of different metabolic groups and ratios (Figs S5 and S6). This observation is consistent with a catabolic switch of cancer cells preincubated with citrate for 48 h associated with increased invasive activity (epithelial-mesenchymal transition [EMT]; Cha et al, 2015; Luo et al, 2017; Wu et al, 2019).

We have also compared the intracellular metabolite levels of PC-3M cells preincubated without or with extracellular 200 μM citrate, with citrate and 150 μM gluconate and 150 μM gluconate alone (Fig S7). A detailed statistical analysis confirmed differences between the groups and indicated significant metabolites (derived from the "variables of importance in projection" [VIP], Fig S7A and B). Significant metabolites included citrate-synthesis related substrates such as glutamine, glutamate, serine as well as total amino acid levels consistent with the catabolic switch.

To check whether pmCiC expressed in cancer cells and in non-cancerous cells differ in the inhibitor profile, we have performed mitotoxicity assay. The mitotoxicity assay indicated that gluconate at the highest concentrations and in the absence of extracellular citrate is a mitochondrial toxin for PC-3M cancer cells (Figs 2B and S8A). This mitotoxic effect was not observed when gluconate was present in the media of benign cells (PNT2-C2; Fig S8B, fibroblasts; Fig S8C, and different groups of CAFs; Fig S8D–G). The differences in the inhibitor profile between exporting and releasing citrate pmCiC

would suggest some alteration in the structure or insertion of the transporter between the two forms of the pmCiC.

We have already established (Mycielska et al, 2018) that cancer cells grown in the presence of extracellular citrate (200 μM) change their metabolism, whereas gluconate is a specific pmCiC blocker. Fig 2C shows that gluconate significantly alters citrate binding to pmCiC. It appears that gluconate binding to pmCiC does not result in a conformational change in pmCiC detectable by Trp fluorescence quenching (Fig S9A, green curve and blue curve). Citrate on the other hand results in Trp quenching, exhibiting a Kd of 16 mM in the absence of gluconate. Elevated Kd compared with the Km values were reported also for other detergent solubilized transporters (Khafizov et al, 2012). The Kd for citrate increases fourfold in the presence of 100 mM gluconate (Fig 2C). The slope of the citrate/gluconate titration suggests a further, likely allosteric, binding event (Fig S9B). Docking of citrate and gluconate to the pmCiC model locates gluconate on top of the central citrate binding site (Fig 2C). At this position, gluconate binding would indeed not result in any Trp quenching, for example, W215 is too far from a sulfur-containing residue such as methionine. Consequently, gluconate might increase the activation threshold while not blocking the citrate binding site during transport in a competitive manner.

### The presence of extracellular citrate affects cancer cell metabolism and induces invasive phenotypes of cancer cells

We next tested how citrate affects markers known to correlate with EMT, which would be consistent with increased tumour invasiveness (Harner-Foreman et al, 2017) and mesenchymal–epithelial transition (MET) associated with metastatic colonisation (Chao et al, 2010). A 48-h preincubation of cancer cells in extracellular citrate resulted in a decreased Glut1, suggesting reduced glucose uptake. Together with the increased expression of the ATP/ADP carrier, this change is consistent with a switch towards OXPHOS (Fig 3A top left panel) reported to be associated with EMT (Ippolito et al, 2016). 2 wk incubation of cancer cells with extracellular citrate compared with control conditions suggests a metabolic switch towards anaerobic glycolysis consistent with proliferation/colonizing step (Faubert et al, 2020; as shown by increased Glut1 expression; Fig 3A top right panel).

Consistent with the acquisition of an invasive character as already shown by the metabolomic data and cytokine release, cancer cells preincubated with citrate for 48 h showed an increased expression of Snail and vimentin (Fig 3A left, bottom, panel; Heldin et al, 2012). The fact that only some increased EMT features were observed is not surprising as PC-3M was derived from a prostate carcinoma metastasis and likely is already invasive. A longer, 2-wk incubation of cancer cells with extracellular citrate resulted in the opposite effect and a decrease of some EMT markers, suggesting changes consistent with MET and the colonisation stage (Fig 3A, right, bottom panel). Scanning electron micrographs revealed that also cell morphology was citrate dependent (Fig S10). A 48 h preincubation of cancer cells with citrate resulted in more polarized, highly invasive, spindle-shaped cells with higher number of single and amoeboid cells known to correlate with the EMT phase (Figs 3B and S10A; Friedl & Wolf, 2003; Morley et al, 2014).

In line with the colonisation step as indicated by the markers (Fig 3A) PC-3M cells preincubated with citrate for 2 wk released mainly

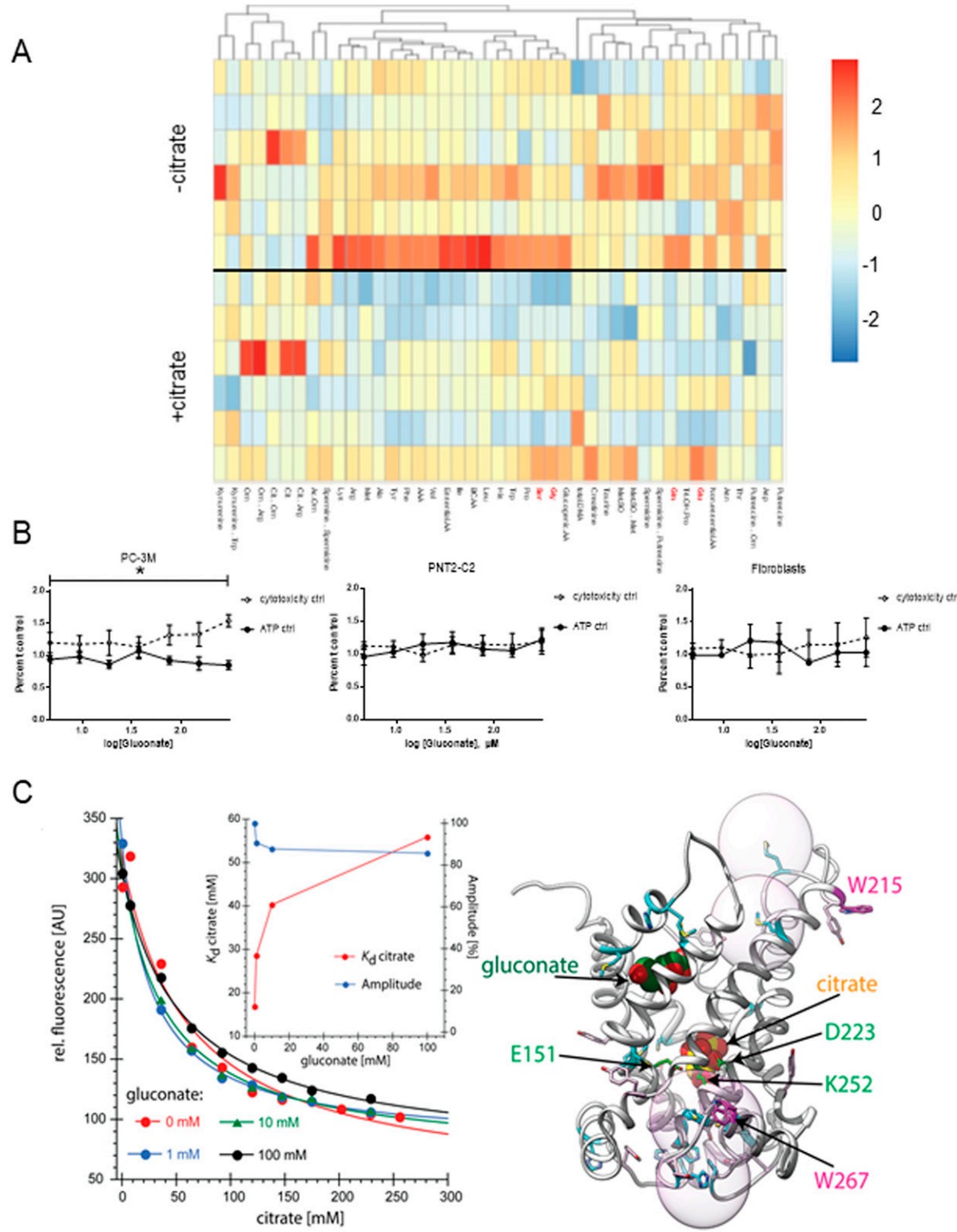

**Figure 2. 48 h of preincubation with extracellular citrate induces intracellular metabolite changes.**
**(A)** Heat map of the intracellular content of amino acids of PC-3M cells grown for 48 h under control conditions or in the media supplemented with 200 $\mu$M citrate. Each row represents a separate repeat. **(B)** Mitotoxicity assay of different types of cells (PC-3M, PNT2-C2, human skin fibroblasts and fibroblasts transformed with conditioned media from PC-3M cells preincubated under control conditions, with extracellular citrate, with gluconate and citrate or with gluconate alone; Fig S7, n ≥ 3) showed that when applied alone at high concentration, gluconate has a cytotoxic effect on cancer cells. **(C)** Trp quenching at different concentrations of citrate and gluconate. Inset: Kd values determined from the Trp quenching (red) and fluorescence change amplitudes (blue) in titrations of citrate in the presence of different concentrations of gluconate. Right panel: Possible inhibition of citrate binding by gluconate binding to a second binding site predicted by a homology model (Mycielska et al, 2018). Spheres indicate Trp quenching radii by sulfur.

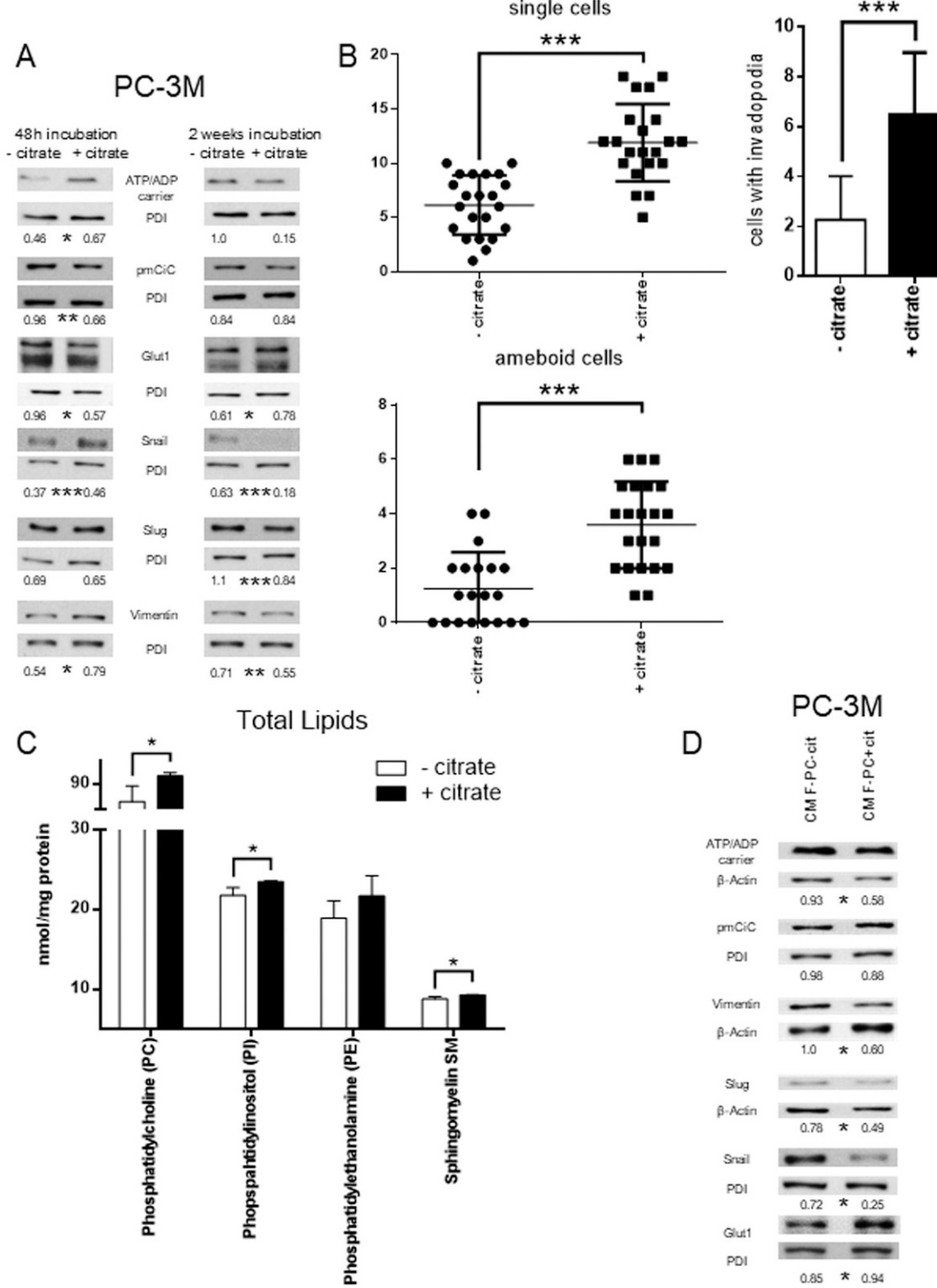

**Figure 3. Citrate in the extracellular space induces metabolic changes and contributes to the metastatic status of cancer cells.**
**(A)** PC-3M cells were preincubated with or without extracellular citrate for either 48 h (left panel) or 2 wk (right panel). Western blots show typical expression of metabolic transporters and EMT/mesenchymal–epithelial transition related proteins (n ≥ 3). The numbers under the Western blots show average normalised values obtained using densitometric analysis; stars depict statistical significance. **(B)** The graphs (left) represent the number of single/ameboid shaped cells in cancer cells preincubated with or without extracellular citrate for 48 h. The graphs on the right hand side show the differences in the number of cells with invadopodia between PC-3M cells grown for 2 wk with or without extracellular citrate. For each condition at least 20 different areas from three repeats were analysed. Pictures were taken with the

cytokines known to support stromal transformation like GROs, IL-8, and IL-3 (Fig S4F). Cytokines released by PC-3M cells kept under control conditions for 2 wk showed a similar pattern as already seen after 48-h preincubation (compare Fig S4A with Fig S4E). This result is consistent with the lack of disease progression/cancer cell reprogramming as already determined by cell morphology and Western blot analysis of cells under control conditions (Fig 3A). Consequently, a similar pattern of cytokine release in CM F-PC–cit after 2 wk and 48 h was also observed (compare Fig S4C with Fig S4G). On the other hand, fibroblasts stimulated with CM PC-3M+cit for 2 wk only released cytokines supporting angiogenesis such as IL-6 and IL-3 (Dentelli et al, 2011; Nagasaki et al, 2014; Masjedi et al, 2018) and metastasis formation/proliferation (MCP1, Loberg et al, 2006) correlating with the colonisation process (Fig S4D and H).

MET occurring under long term incubation of cancer cells with extracellular citrate would also agree with the decreased (but not significantly) expression of the ATP/ADP carrier and increased Glut1 level suggesting an increase in glycolysis (Fig 3A). Increased glycolysis and lactate release could enhance proliferation of the colonizing cells (De la Cruz-López et al, 2019). 2-wk preincubation with extracellular citrate resulted in highly spread/flattened cells with long invadopodia, contacting the ECM (extracellular matrix) and other cells consistent with the colonisation step (Figs 3B and S10B; Williams et al, 2019). Importantly, under controlled conditions, cellular morphology remained unchanged at all times. The morphology of cancer cells grown with citrate as depicted with scanning electron microscopy seems also to indicate increased fluidity of their plasma membrane. Consistently, the level of total phosphatidylcholine, phosphatidylinositol, sphingomyelin, and phosphatidylethanolamine (although the last not significant) was increased in cancer cells grown with citrate (Dawaliby et al, 2016; Fig 3C).

We have already shown that citrate appears only in the media from CAFs and is not present at any significant level in the conditioned media from cancer cells. However, conditioned media from cancer cells contained significant amounts of cytokines, known to support aggressive cancer behaviour. To test whether cancer cell-derived cytokines and potentially other substrates are sufficient to induce changes in cancer cells, we preincubated PC-3M cells with CM PC-3M+cit or CM PC-3M–cit. We did not observe any significant and consistent changes between the two groups of cancer cells (Fig S11A). However, incubation of PC-3M cells with CM F-PC+cit or CM F-PC–cit resulted in more pronounced differences in cancer cells (Fig 3D). These results indicate that citrate released by CAFs together with other elements might play a crucial role in supporting cancer progression.

The data above show that stromal transformation depends on the metabolic status of cancer cells. CAFs compensate for the lack of metabolites and cytokines in cancer cells to support their invasive and metastatic behaviour. From these data, we proposed that citrate availability is an important requirement for tumour progression and this was investigated further below.

## Inhibition of pmCiC with gluconate in vivo decreases metastatic spread and stromal transformation

To test the role of extracellular citrate uptake in metastasis, we injected the lower spleen pole of athymic nude mice with human pancreatic cancer L3.6pl cells. To avoid side effects of intrasplenic "primary tumours," the spleen was removed 15 min after tumour cell injection. Gluconate blockade (daily in vivo injection) of citrate uptake by metastasising human pancreatic cancer cells significantly reduced metastasis rate (Fig 4A). Reduced metastatic spread was accompanied by decreased stromal transformation ($\alpha$SMA) as shown in Fig 4B. Additional controls showed that gluconate or citrate alone does not induce any changes in the $\alpha$SMA expression in fibroblasts grown under standard conditions (Fig S11B). Staining with the anti-fibroblast–activating protein (FAP) antibody showed a pronounced transformation of some stromal cells consistent with hepatic stellate cells in the control group (Fig 4C; Wang et al, 2005; Kaps & Schuppan, 2020). This was not observed in the case of gluconate treated mice. Moreover, the size of metastases in the gluconate-treated group was on average smaller with increased immune infiltration (Fig 4D). Although FAP is not a typical immune cell marker, its expression has been previously reported in sub-populations of CD45+/CD45– cells (Jiang et al, 2016).

Treatment of mice injected subcutaneously with human pancreatic L3.6pl cells reduced tumour growth and more importantly changed the metabolic characteristics of cancer tissues (Mycielska et al, 2018). Changes in the tumour tissues were already visible from the beginning of the experiment (Fig 5A, details in Mycielska et al [2018]). Experimental tumours in mice treated with gluconate were smaller, rounder, and paler compared to their untreated counterparts. This observation is consistent with decreased angiogenesis (as shown by CD31 staining Fig 5B) accompanied by a significant reduction of stromal transformation ($\alpha$-SMA; Fig 5B). There was also an increase in apoptosis which was most pronounced at the bottom of the gluconate-treated tissues as determined by TUNEL staining (Fig 5C). In mice treated with gluconate, there was an overall decrease in the expression of PDGFR$\beta$ and vimentin (Fig 5D) particularly at the tumour–stroma interface. Decreased levels of PDGFR$\beta$ and vimentin in non-cancerous tissue are also consistent with decreased stroma transformation (Fig 5C).

We have also performed a CAM assay (chorioallantoic membrane) used as a 3D in vivo tumour model. Treatment of L3.6pl cells with Na$^+$-gluconate significantly reduced tumour growth and volume (Fig 5E) as compared with the control conditions where equivalent volume of NaCl was used.

---

scanning electron microscope. **(C)** Measurement of phosphatidylcholine (PC), phopspahtidylinositol (PI), phosphatdylethanolamine (PE), and sphingomyelin (SM) in cells under experimental conditions with or without extracellular citrate for 48 h (n = 4). **(D)** Western blot analysis of the expression of metabolic transporters and EMT/mesenchymal–epithelial transition–related proteins of PC-3M cells grown for 48 h in the media from fibroblasts transformed with the conditioned media from PC-3M cells preincubated for 48 h with or without extracellular citrate (CM F-PC–cit and CM F-PC+cit, respectively; right panel; n = 3). The numbers under the Western blots show average normalised values obtained using densitometric analysis; stars depict statistical significance.

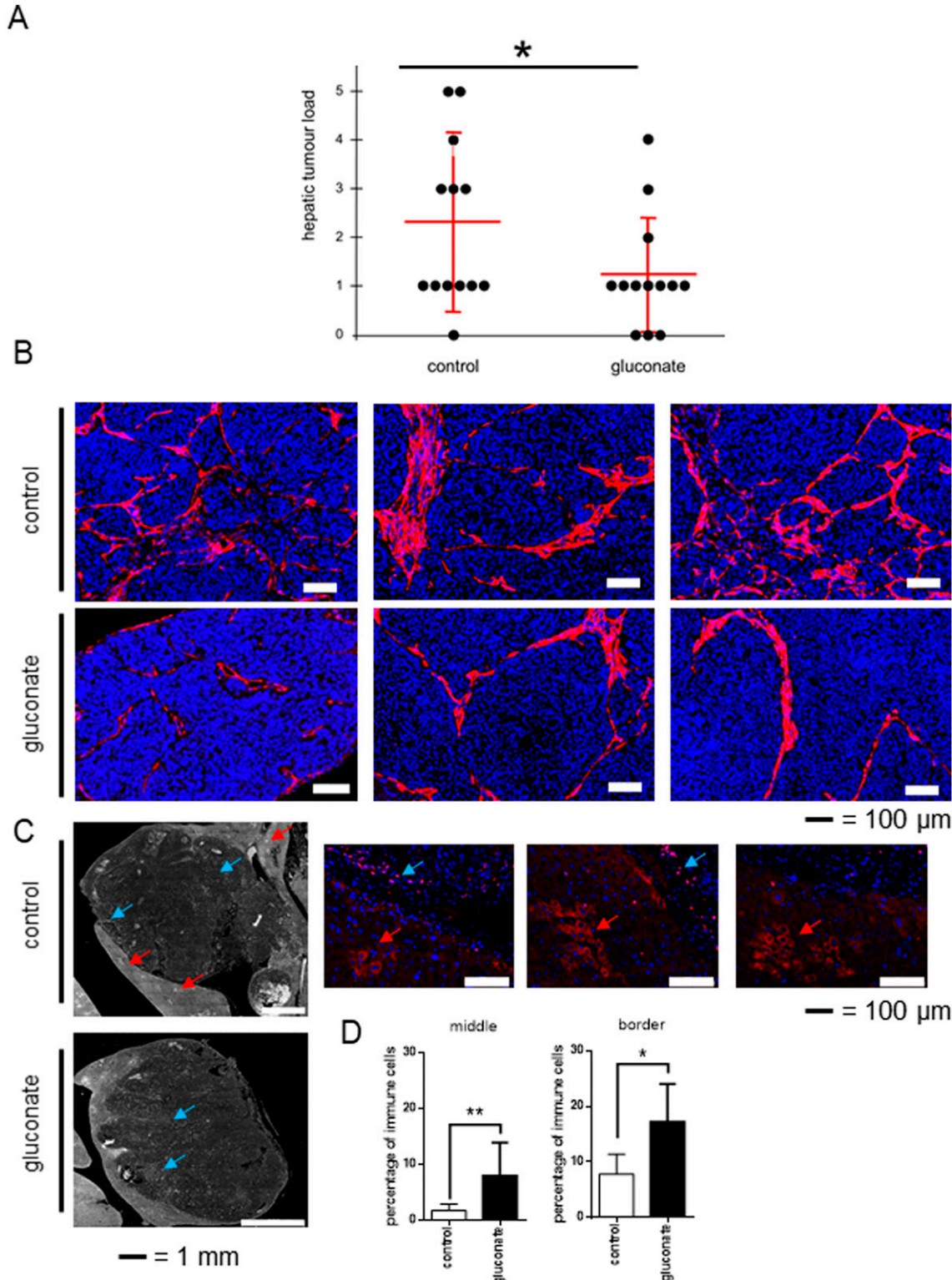

**Figure 4. Blocking of citrate uptake by cancer cells decreases metastasis rate in vivo.**
Human pancreatic cancer L3.6pl cells were injected into the lower spleen pole of immunodeficient mice. 15 min after tumour cell injection, the spleen was removed. Animals were daily intraperitoneally injected with sodium gluconate (500 mg/kg/d), respectively, NaCl. After 24 d, mice were euthanized, and livers and laparotomy wounds were assessed. **(A)** Liver metastases were evaluated by a macroscopic score (0 = no tumour load; 1 = small singular tumours; 2 = large singular tumours; 3 = confluent tumours in less than half of liver; 4 = confluent tumours in more than half of the liver; 5 = tumours in all liver segments). Hepatic tumour load was significantly decreased ($P = 0.046$; n = 13 mice per group, one-tailed unpaired $t$ test) in the treatment group (500 mg/kg/d sodium gluconate) compared with control group (NaCl).

To conclude, depriving cancer cells of citrate by applying gluconate in vivo reduced the cancer growth and metastasis rate, angiogenesis, and stromal transformation. Hence, blocking citrate uptake in vivo might result in reduced capability of tumour to progress.

### pmCiC expression in cancer-associated cells in human cancerous tissues

We next tested whether our in vitro data indicating that CAFs release citrate to support cancer metabolism can be extended to the in vivo environment, by immunostaining analysis of human tissue (Fig 6A–F). Analysis of human pancreatic (Fig 6A and C) and gastric cancer glands (Fig 6B and D) as well as lymph node metastasis (Fig 6E) and lung metastasis (Fig 6F) of gastric cancer confirmed that pmCiC is expressed not only in cancer cells of different organs (Mycielska et al, 2018) but also in some subsets of fibroblasts in the surrounding tumour environment. Moreover, we have also found pmCiC expressed in some of the tumour micro-vessels. This would be consistent with extracellular citrate playing a role in angiogenesis by supporting vessel formation. To complete these studies, we have assessed pmCiC expression in cancer cells and cancer-associated stromal cells of human breast, gastric and pancreatic tissues (Fig 6G). The highest expression of pmCiC in stromal cells was observed in gastric and pancreatic cancers known to intensively produce desmoplastic stroma as compared with other cancer types (Hosein et al, 2020; Oya et al, 2020). Therefore, targeting pmCiC might have an effect not only on cancer cells but also on the tumour's "infrastructure" (supporting cells).

# Discussion

We have previously shown that cancer cells of different origin take up extracellular citrate through the recently cloned pmCiC (Mycielska et al, 2018). In the present study, we have explored the source of citrate used by cancer cells and, importantly, its potential role in tumour progression.

We have determined that CAFs, together with other cells from the cancer environment such as hepatic stellate cells, express pmCiC with the function of citrate release (summarised in Fig S12A and B). Increased citrate synthesis by cancer-associated stroma (CAS) is consistent with a recent report showing increased glutamine uptake by cancer-associated cells rather than by cancer cells themselves in vivo (Marin-Valencia et al, 2012).

Our data show that the metabolic activity of CAFs depends on extracellular citrate availability to cancer cells; therefore, it is regulated in the cancer environment. We propose that a lack of extracellular citrate stimulates cancer cells to transform their surroundings to get the necessary substrate. In support of this, we

show that cancer cells deprived of extracellular citrate for 48 h release IL-6, which stimulates angiogenesis and GROs involved in stromal transformation/senescence (Nagasaki et al, 2014; Masjedi et al, 2018). Indeed, senescent fibroblasts which are known to be part of the cancer environment (Yang et al, 2006; Mellone et al, 2017), have previously been shown to release high levels of citrate (James et al, 2015). Therefore, cancer cells deprived of citrate may be selected to release factors to stimulate their surroundings to compensate for citrate deficiency. The organ colonisation step, as seen after 2 wk of incubation with extracellular citrate, was accompanied by a release of IL-3, an angiogenesis-stimulating cytokine, complemented by GROs responsible for the stromal transformation. Cancer cells grown long term without citrate released similar cytokines at 48 h and 2 wk, corroborating the hypothesis that citrate is necessary for metastatic reprogramming. Previous studies have shown that other metabolites such as lactate can also play a role in the communication between cancer cells and CAFs (Whitaker-Menezes et al, 2011), but there is no evidence that lactate alone can modify tumour behaviour. Conversely, lactate can also be used for citrate synthesis (reviewed by Mycielska et al [2015] and Haferkamp et al [2020]).

Persistent and long-term presence of citrate in the media of cancer cells was shown to induce MET associated with growth and colonisation (Brabletz, 2012). This would be consistent with extracellular citrate playing an additional signalling role, where the constant presence of citrate might complete the process of stromal transformation creating a niche ready for the metastatic growth. Previous studies have shown that increased citrate levels precede leptomeningeal carcinomatosis from lung cancer (An et al, 2015), suggesting that cancer cells might transform distant stroma and induce local citrate increase before metastatic processes begins.

The shorter incubation time of cancer cells with extracellular citrate was shown to induce an invasive phenotype with some features consistent with EMT. A similar effect on the protein level was obtained when the cells were incubated with the media from fibroblasts transformed by cancer cells deprived of citrate, therefore induced to supplement this metabolite. This supplementation of citrate by CAS and the induction of an activated fibroblast phenotype could have several implications regarding potential cancer therapies. It is possible that the need for de novo stromal transformation (e.g., after chemotherapy) could induce metastatic processes (EMT and MET) through the local changes of metabolite and cytokine levels and therefore modify the metastatic behaviour of the remaining cancer cells.

Incubation of cancer cells with extracellular citrate decreased an overall level of the metabolites tested. In particular, there was a decrease of intracellular levels of substrates known to be involved in intracellular citrate synthesis, including serine and glycine (Possemato et al, 2011), glutamine and glutamate (Metallo et al, 2011). This would suggest that cancer cells without extracellular citrate supply are forced to use several additional pathways to

---

**(B)** Pictures showing differences in the αSMA staining pattern in metastasis in mouse livers treated with gluconate versus control. **(C)** Liver metastasis were stained with the anti-fibroblast activating protein (FAP) antibody. Red arrows show stained area in the stroma consistent with cancer-associated cells, whereas blue arrows depict immune cells expressing FAP. Enlarged pictures from the control group (on the right) show cells stained with anti-FAP antibody (in red) in the stroma (indicated with red arrows) and immune cells (blue arrows). **(D)** Graphs showing average number of immune cells per total number of cells in the studied area. For these measurements two border and two central areas were calculated per metastasis. For each experimental group, four different metastases were evaluated.

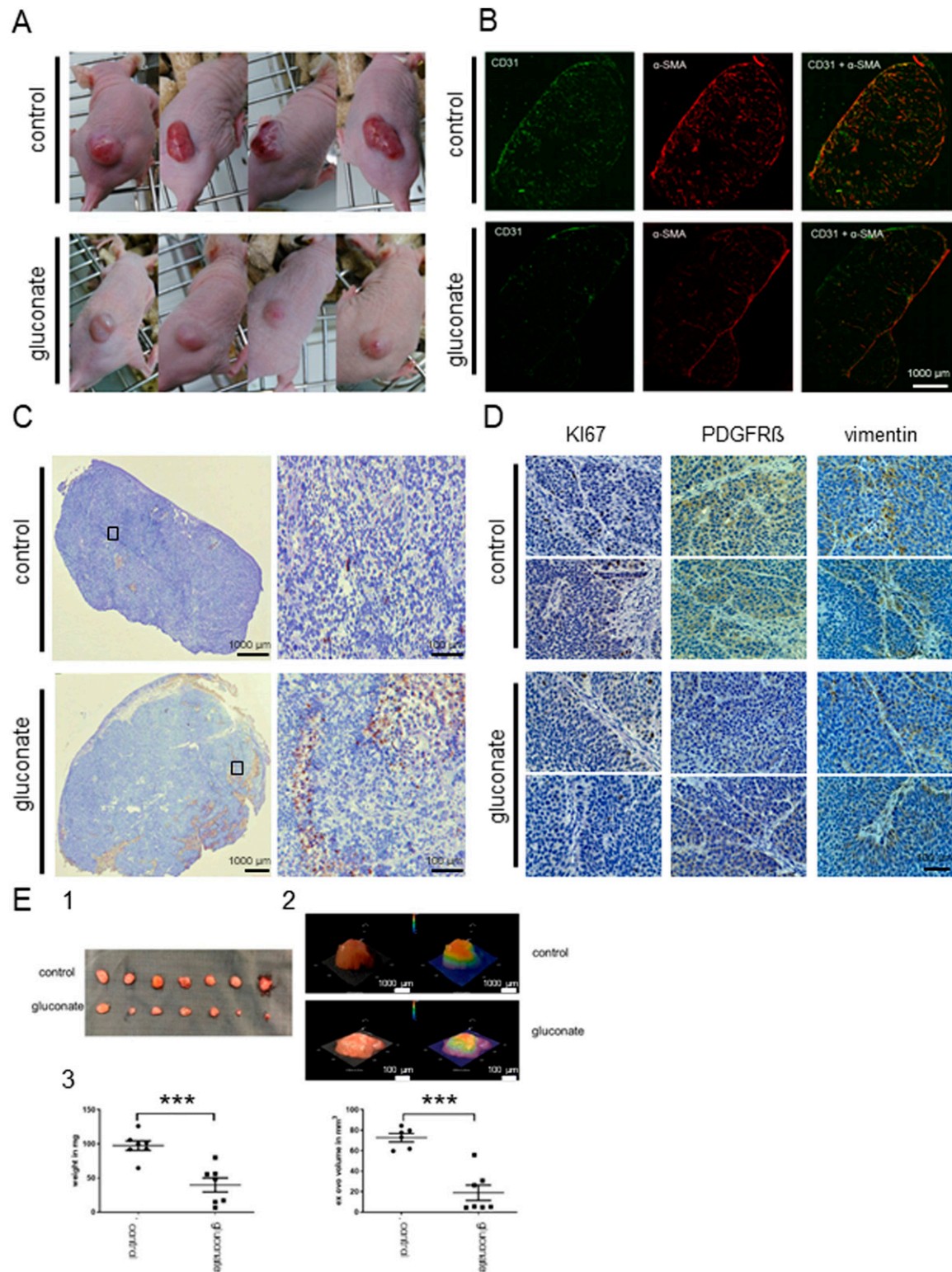

**Figure 5. Gluconate treatment in vivo increases apoptosis, decreases angiogenesis, stromal transformation, proliferation and expression of PDGRβ and vimentin in cancerous tissues.**

Human pancreatic cancer cells L3.6pl were injected subcutaneously in nude mice. Control mice were injected with saline and treated group with Na⁺-gluconate (as described before, Mycielska et al, 2018). **(A)** Pictures of mice showing qualitative differences between the cancerous growths in treated versus untreated animals. The photos cannot be used to compare tumour size. **(B)** Angiogenesis (CD31) and stromal transformation (αSMA) were also significantly decreased in the case of treated animals. **(C)** Staining of the sections of tumours (TUNEL) show increased apoptosis in the treated group. Moreover, apoptotic regions were mainly observed at the lower

account for the missing metabolite. Availability of extracellular citrate decreases the levels of several metabolites which can be considered as by-products of increased citrate synthesis (Haferkamp et al, 2020). This metabolic change and more balanced use of metabolic pathways are likely to contribute to the acquisition of a more aggressive phenotype.

Although gluconate had some mitotoxic effects on cancer cells when applied alone, it has to be noted that this effect was observed in the presence of the highest concentrations of gluconate only, much higher than those applied in our in vivo and in vitro experiments. It can be therefore deduced that the effects of gluconate observed in the experiments are due to its specific action on pmCiC and not through some less specific mitochondrial blocking. Moreover, gluconate, till now, has been considered as a physiologically neutral substance and used in medicine as a heavy ion carrier (reviewed by Mycielska et al [2019]). It is also used in electrophysiology and transport studies as a Cl⁻ replacement because it is plasma membrane impermeable (Carini et al, 1997; Bonzanni et al, 2020).

Importantly, we have confirmed that depriving cancer cells of extracellular citrate supply in vivo results in decreased metastatic spread, stromal transformation and angiogenesis. Furthermore, we have observed increased immune cell infiltration of the mouse tumour xenografts treated with gluconate. FAP expressing immune cells have been seen previously (Arnold et al, 2014; Cremasco et al, 2018) and considered to comprise CD45⁺ cells, which is consistent with the increased proinflammatory cytokine release by cancer cells deprived of extracellular citrate. Increased release of proinflammatory cytokines has been also associated with citrate uptake by monocytes (Ashbrook et al, 2015), whereas activated macrophages have been determined to accumulate citrate (Jha et al, 2015). This could suggest that preventing cancer cells from taking up extracellular citrate and consequently increasing local citrate concentration might have a stimulating effect on anti-cancer immune response. Although we found these results particularly interesting, it has to be acknowledged that for this study, immunocompromised mice have been used, and therefore, a more detailed study of immune activity upon gluconate application was not possible.

To conclude, our results demonstrate that citrate is an important metabolite supportive of tumour progression which is synthesized and released by CAS. Moreover, citrate synthesis and release from cancer-associated stroma appears to be one of their major metabolic tasks and is induced and controlled by cancer cells through cross-cellular interactions. We have identified the transporter responsible for citrate release and confirmed its expression in cancer-associated cells in human tissues. Blocking of citrate uptake by cancer cells using gluconate, a specific inhibitor of citrate uptake, reduced cancer spread and decreased stromal transformation and angiogenesis in vivo.

Our novel finding of citrate release by cancer-associated stroma is crucial in understanding the mechanism of tumour environmental support and interactions between cancer cells and the

environment. It sheds new light on the mechanism of how stroma stimulates metastatic spread, facilitates organ colonisation and angiogenesis, as well as increases cancer cells' resistance to anti-cancer therapies. Inhibiting citrate release from CAS and/or uptake by cancer cells could offer novel, specific, and readily implemented options for cancer treatment and metastasis prevention.

# Materials and Methods

### Cell culture

Cell lines PC-3M, PNT2-C2, and L3.6pl were grown as described previously (Mycielska et al, 2018). Adult human dermal primary fibroblasts were derived from the skin biopsies of healthy donors as described by Milenkovic et al (2018). Naive human hepatic stellate cell line hHSC (iX Cells Biotechnologies) and two permanently activated human hepatic stellate cell lines LX2 and HSC^hTERT (Meyer et al, 2017) were cultured as depicted before (Schmidt et al, 2018). The following chemicals were used: citric acid and Na⁺-gluconate (Sigma-Aldrich), and dialyzed serum (PAN Biotech GmbH). The following antibodies were used: pmCiC antibodies (Mazurek et al, 2010; custom-made by GenScript Inc.), SLC25A5, vimentin, Slug, Snail, EGFR, E-cadherin, N-cadherin (Cell Signalling), SLC6A14, Glut1, Ki67, pan cytokeratin (Abcam), TGFβ-RII, PDGFRα (Santa Cruz Biotechnology), and TUNEL (R&D Systems). Experimental media consisted of glucose-free Roswell Park Memorial Institute 1640 medium (RPMI-1640) (Lonza), 5% dialyzed serum, 2 mM glutamine, 0.5 g/l glucose, and ±200 µM citrate, unless otherwise stated. The incubation time varied between 48 h, 2 wk, and over 2 mo, as specified. Western blots were analysed by measurement of the pixel density using ImageJ software (Laboratory for Optical and Computational Instrumentation [LOCI], University of Wisconsin-Madison).

For conditioning, the cells were plated at $1 \times 10^6$ (PC-3M) or $2 \times 10^6$ (L3.6pl) per T75 flask in regular growth media for 24 h (PC-3M) or 48 h (L3.6pl). After that, they were incubated in RPMI or DMEM, respectively, with 0.5% FBS and 2 mM glutamine for 48 h (Fig S1). Media were then collected, filtered, and supplemented with 2 g/l (PC-3M) or 4 g/l (L3.6pl) glucose and 2 mM glutamine to avoid starvation. Fibroblasts were grown in conditioned media for 72 h and collected for further analysis. To study the extracellular citrate effect on fibroblast activation, PC-3M cells were preincubated in RPMI media with 5% dialyzed serum, 2 mM glutamine, 0.5 g/l glucose, and with or without 200 µM citrate. After 48 h, media were changed to conditioning media and the procedures were carried out as described above. Hepatic stellate cells were exposed to regular medium (10% FBS), low serum medium (1% FBS), and conditioned medium from PC-3M and L3.6pl for 48 h (LX2, HSC^hTERT) or 72 h (hHSC); afterwards cells were harvested and pmCIC expression was measured by Western blot.

---

parts of the tissues. **(D)** Cancer cells proliferation in the tissues was decreased (Ki67), as well as PDGFRβ and vimentin expression. **(E)** The CAM assay was performed using L3.6pl cells. The cells were left to grow for 2 d then NaCl was used in the control group and Na⁺-gluconate in gluconate group for 5 d. (1) Tumour explants after daily treatment with NaCl (upper) or Na⁺-gluconate (lower). (2) Examples of tumour volume measurements with the 3D digital microscope (Keyence VHX-7000 microscope). (3) Graphs showing changes in the tumour weight and tumour volume of NaCl versus gluconate treated groups.

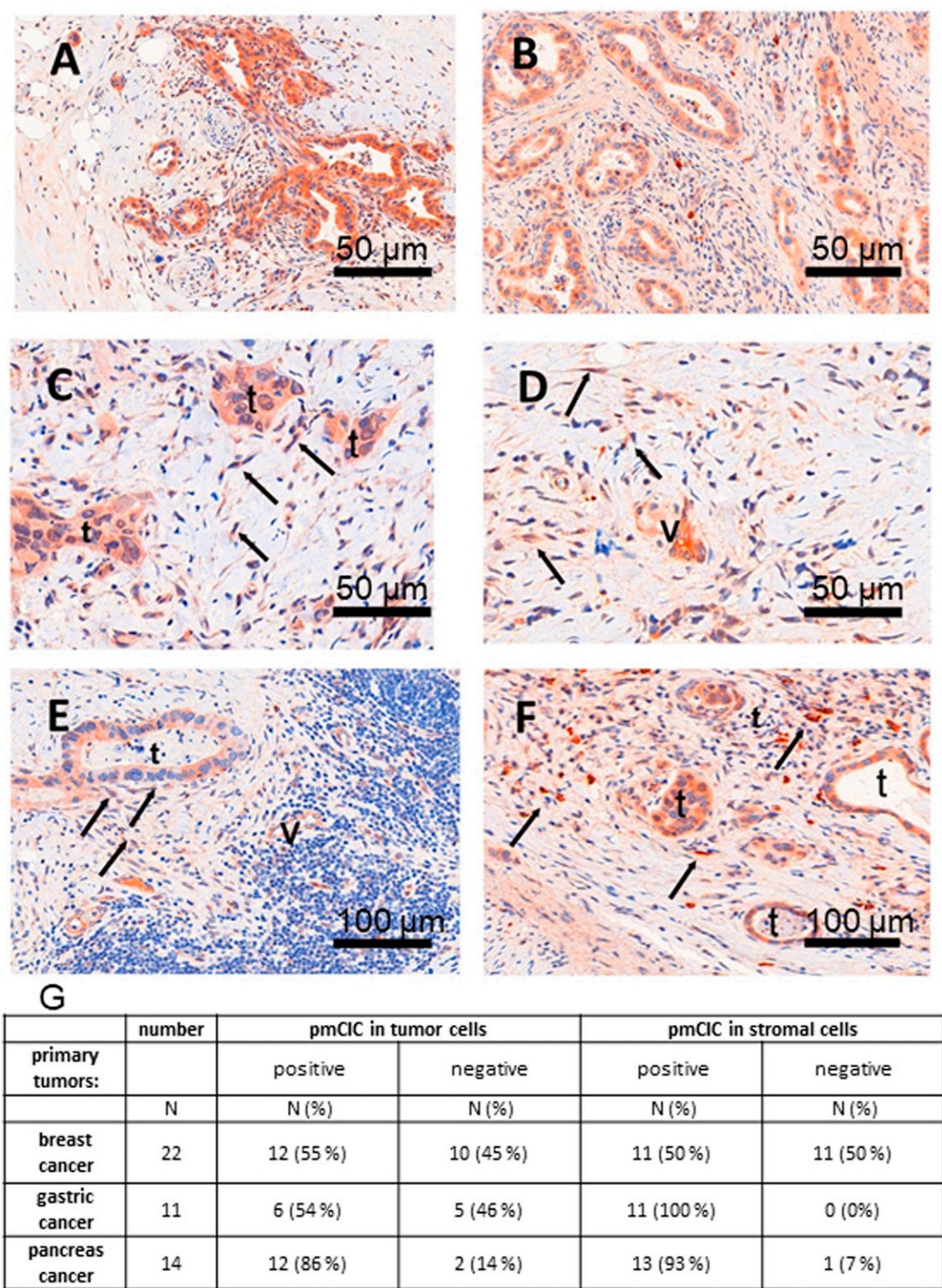

**Figure 6. pmCiC is expressed in the stroma of human cancerous tissues.**
**(A, B)** immunohistochemical staining of pmCIC showed strong expression in human pancreatic (A) and gastric cancer glands (B) (DAB—150 $\mu$m). **(C, D)** In close contact to infiltrating cancer glands, there is also a prominent expression in micro-vessels (v) but also in spindle cells (→) of the tumour microenvironment (TME) in both cancer types (C, D) (DAB, — 50 $\mu$m). The latter cell type of the TME could be identified by vimentin as tumour associated fibroblasts (data not shown). **(E)** Consecutive examination of lymph node metastasis (E) (DAB—50 $\mu$m) demonstrate a positive tumour gland but also expression in micro-vessels (v) and fibroblasts. **(F)** The same was true for a metastatic gastric cancer shown in figure (F) (DAB, — 50 $\mu$m). **(G)** A table showing the number/percentage of human tumour tissues expressing pmCiC in cancer cells and in cancer-associated stroma cells.

## Seahorse assay

$3 \times 10^4$ primary dermal fibroblasts cells were grown in XFp eight well miniplates (Agilent Technologies) at 37°C, humidified air, and 5% $CO_2$/95% air. Cartridges were prepared according to the manufacture's recommendations. The XFp Cell Mito Stress Test Kit (Agilent Technologies) contained the mitochondrial stress compounds oligomycin (1 $\mu M$), FCCP (2 $\mu M$), and rotenone/antimycin A (1 $\mu M$). Oxygen consumption rate and extracellular acidification rate (ECAR) were measured by means of an XFp Seahorse Flux Analyzer (Agilent Technologies) and were normalised by cell number. Directly after measurement, the cells were fixed and stained with Hoechst 33342. The cell number per well were counted using *ImageJ* software.

## Cytokine detection and citrate measurements

To measure the release of cytokines, Human Cytokine Antibody Array from RayBiotech was used. Conditioned media were collected from cancer cells as summarised in Fig S3. Cytokine values from the unconditioned media used for conditioning were subtracted from the values obtained in the media from cancer cells. Consequently, cytokine levels in the media from cancer cells were subtracted from the values obtained in the media from fibroblasts. The kit was used according to the manufacturer's instructions and analysed using supplied positive and negative controls for all the assays. We used the same control media for all conditions tested, and all the membranes were prepared at the same time. The values between different cell types were not compared, but the trend in cytokines released is presented on separate graphs for all conditions tested.

Citrate in the media from fibroblasts was measured using Citrate assay kit (Sigma-Aldrich) according to the manufacturer instructions.

## Inhibition of citrate binding to pmCiC analysed by fluorescence quenching

pmCiC was expressed in Sf9 cells using the Baculo-Virus System. Membranes were prepared by sequential centrifugation and solubilized with 2% vol/vol dodecylmaltoside. pmCiC was purified via His-Tag Ni-NTA chromatography and brought to a final concentration of 0.1 mg/ml for fluorescence quenching titration experiments using a JASCO Cary Eclipse spectrometer. The excitation wavelength was set to 285 nm, and emission was measured in between 200 and 500 nm. Trp quenching was detected at different concentrations of citrate in the absence and presence of gluconate. Relative changes in the fluorescence amplitude at 375 nm was plotted against citrate or gluconate concentration and fitted with Hill equation. The double reciprocal plot of fluorescence change against citrate concentration was fitted with a non-linear model. Kd values were determined from rectangular hyperbolic binding models and plotted against respective gluconate concentrations. Amplitude, that is, the maximal change of fluorescence during citrate titration was plotted against respective gluconate concentrations. A previously constructed homology model of pmCiC with citrate and gluconate docked to it (Mycielska et al, 2018) was investigated by for putative TRP quenching radii by sulfur using the program UCSF Chimera (Mycielska et al, 2018).

## Non-targeted metabolomics

Non-targeted metabolomics analysis of the cell media was conducted at the Genome Analysis Center, Research Unit Molecular Endocrinology and Metabolism, Helmholtz Zentrum München. The samples were stored at −80°C before analysis. On the day of extraction, the samples were thawed on ice and were randomized before 100 $\mu l$ of each sample was pipetted into a well in a 2-ml 96-well plate. In addition to samples from this study, a pool of samples of the study was aliquoted into aliquots of 100 $\mu l$. The aliquots were distributed in six wells of the 96-well plate and were extracted as the samples of the study. Besides those samples, 100 $\mu l$ of human reference plasma sample (Seralab) was also extracted as the samples of the study. These samples served as technical replicates throughout the data set to assess process variability. In addition, 100 $\mu l$ of water was extracted as samples of the study and placed in six wells of each 96-well plate to serve as process blanks.

Protein was precipitated and the metabolites in the samples were extracted with 475 $\mu l$ methanol, containing four recovery standard compounds to monitor the extraction efficiency. After centrifugation, the supernatant was split into four aliquots of 100 $\mu l$ each onto two 96-well microplates. The first two aliquots were used for LC–MS/MS analysis in positive and negative electrospray ionization mode. Two further aliquots on the second plate were kept as a reserve. The samples were dried on a TurboVap 96 (Zymark, Sotax). Before LC–MS/MS in positive ion mode, the samples were reconstituted with 50 $\mu l$ of 0.1% formic acid and those analysed in negative ion mode with 50 $\mu l$ of 6.5 mM ammonium bicarbonate, pH 8.0. Reconstitution solvents for both ionization modes contained further internal standards that allowed monitoring of instrument performance and also served as retention reference markers. To minimize human error, liquid handling was performed on a Hamilton Microlab STAR robot (Hamilton Bonaduz AG).

LC–MS/MS analysis was performed on a linear ion trap LTQ XL mass spectrometer (Thermo Fisher Scientific GmbH) coupled with a Waters Acquity UPLC system (Waters GmbH). Two separate columns (2.1 × 100 mm Waters BEH C18 1.7 $\mu m$ particle) were used for acidic (solvent A: 0.1% formic acid in water, solvent B: 0.1% formic acid in methanol) and for basic (A: 6.5 mM ammonium bicarbonate, pH 8.0, B: 6.5 mM ammonium bicarbonate in 95% methanol) mobile phase conditions, optimized for positive and negative electrospray ionization, respectively. After injection of the sample extracts, the columns were developed in a gradient of 99.5% A to 98% B in 11 min run time at 350 $\mu l$/min flow rate. The eluent flow was directly connected to the ESI source of the LTQ XL mass spectrometer. Full-scan mass spectra (80–1,000 m/z) and data-dependent MS/MS scans with dynamic exclusion were recorded in turns. Metabolites were annotated by curation of the LC–MS/MS data against proprietary Metabolon's chemical database library (Metabolon, Inc.) based on retention index, precursor mass, and MS/MS spectra. In this study, 109 metabolites, 63 compounds of known identity (named biochemical), and 46 compounds of unknown structural identity (unnamed biochemical) were identified. The unknown chemicals are indicated by a letter X followed by a number as the compound identifier. The metabolites were assigned to cellular pathways based on PubChem, KEGG, and the Human Metabolome Database.

## Cell collection, homogenization, and targeted metabolomics

The medium was aspirated, the PC-3M cells were quickly washed twice with 2 ml warm PBS, and their metabolism was subsequently quenched by the addition of pre-cooled (dry ice) 300 $\mu l$ extraction solvent, a 80/20 (vol/vol) methanol/water mixture. Cells were scraped off the culture vessel using rubber tipped cell scrapers (Sarstedt) and together with the solvent collected in pre-cooled micro tubes (0.5 ml; Sarstedt). The culture well was rinsed with another 100 $\mu l$ extraction solvent, and the liquid was also transferred to the tube. The samples were stored at −80°C until further use.

For homogenization, 80-mg glass beads (0.5 mm, VK-05; PeqLab) were added to the cell samples, which were homogenized using the Precellys24 homogenizer at 0–3°C for two times over 25 s at 3,720$g$ with a 5-s pause interval.

To normalize the obtained metabolomics data from cell homogenates for differences in cell number, the DNA content was determined using a fluorescence-based assay for DNA quantification. The assay was performed as previously described (Muschet et al, 2016). Briefly, the fluorochrome Hoechst 33342 (10 mg/ml in water; Life Technologies, Thermo Fisher Scientific) was diluted in PBS to the final concentration of 20 $\mu g$/ml. 80 $\mu l$ of this dilution was applied to each well of a black 96-well plate (F96, Nunc; Thermo Fisher Scientific). After brief vortexing of the cell homogenates, 20 $\mu l$ of the sample was added to the Hoechst 33342 dilution to gain 100 $\mu l$ of total volume per well and mixed by pipetting. Each sample was applied to the plate in four replicates. 20 $\mu l$ extraction solvent was used for blank measurements. The plate was incubated at room temperature in the dark for 30 min and the fluorescence was read using a GloMax Multi Detection System (Promega) equipped with an UV filter ($\lambda$Ex 365 nm, $\lambda$Em 410–460 nm; Promega). Subsequently, the samples were centrifuged at 4°C and 11,000$g$ for 5 min, and 10 $\mu l$ of the supernatant was used for the metabolite quantification.

The targeted metabolomics approach was based on liquid chromatography-electrospray ionization-tandem mass spectrometry (LC-ESI-MS/MS) and flow injection-electrospray ionization-tandem mass spectrometry (FIA-ESI-MS/MS) measurements by AbsoluteIDQ p180 Kit (BIOCRATES Life Sciences AG). The assay allows simultaneous quantification of 188 metabolites out of 10 $\mu l$ plasma and includes free carnitine, 39 acylcarnitines (Cx:y), 21 amino acids (19 proteinogenic + citrulline + ornithine), 21 biogenic amines, hexoses (sum of hexoses–about 90–95% glucose), 90 glycerophospholipids (14 lysophosphatidylcholines [lysoPC] and 76 phosphatidylcholines [PC]), and 15 sphingolipids (SMx:y). The abbreviations Cx:y are used to describe the total number of carbons and double bonds of all chains, respectively (for more details see Römisch-Margl et al (2012)). For the LC-part, compound identification and quantification were based on scheduled multiple reaction monitoring measurements (sMRM). The method of AbsoluteIDQ p180 Kit has been proven to be in conformance with the EMEA-Guideline "Guideline on bioanalytical method validation (21 July, 2011)," which implies proof of reproducibility within a given error range. Sample preparation and LC–MS/MS measurements were performed as described in the manufacturer in manual UM-P180. Analytical specifications for limit of detection (LOD) and evaluated quantification ranges, further LOD for semiquantitative measurements, identities of quantitative and semiquantitative metabolites, specificity, potential interferences, linearity, precision and accuracy, reproducibility, and stability were described in Biocrates manual AS-P180.

The LODs were set to three times the values of the zero samples (extraction solvent). The LLOQ and ULOQ were determined experimentally by Biocrates.

The assay procedures of the AbsoluteIDQ p180 Kit as well as the metabolite nomenclature have been described in detail previously (Römisch-Margl et al, 2012; Zukunft et al, 2013).

Sample handling was performed by a Hamilton Microlab STAR robot (Hamilton Bonaduz AG) and an Ultravap nitrogen evaporator (Porvair Sciences), beside standard laboratory equipment. Mass spectrometric analyses were carried out on an API 4000 triple quadrupole system (Sciex Deutschland GmbH) equipped with a 1200 Series HPLC (Agilent Technologies Deutschland GmbH) and a HTC PAL auto sampler (CTC Analytics) controlled by the software Analyst 1.6. Data evaluation for quantification of metabolite concentrations and quality assessment was performed with the software MultiQuant 3.0.1 (Sciex) and the MetIDQ software package, which is an integral part of the AbsoluteIDQ Kit. Metabolite concentrations were calculated using internal standards and reported in micrometres.

In addition to the investigated samples, five aliquots of a pooled reference plasma (Ref_Plasma-Hum_PK3) were analysed on each kit plate. The results of these reference plasma aliquots can be used for calculation of potential batch effects and data normalization (of different studies).

## Mitochondrial toxicity test

Mitochondrial ToxGlo Assay by Promega was used. Cells were preincubated with either control medium or medium containing 200 $\mu M$ citrate for 48 h. The assay was performed according to the manufacturer's instructions. As recommended, galactose was used instead of glucose. Highest citrate concentrations were set to be at 10 mM, highest gluconate concentrations were set to be at 600 $\mu M$.

## Immunohistochemistry

Human tissue (use granted by the Ethics Commission of the University of Regensburg, number 14-101-0263) was stained with pmCiC specific antibody, as described before (Mazurek et al, 2010). Expression of the pmCiC in cancer cells and cancer-associated stroma was assessed by scanning up to 20 high-power fields from one slide (Zeiss Microscope Axiovision). The total number of positive stained tumour cells was counted and depicted in percentage of all tumour cells. The same method was applied for the assessment of stromal cells.

To calculate the relative number of the immune cells in liver metastasis, we have used ImageJ software. Two border and two middle areas of 250 $\mu m$ by 250 $\mu m$ were evaluated for each metastasis (four per experimental group). For each area the number of nuclei and immune cells were calculated. The results were expressed as a ratio of the number of immune cells to the total number of nuclei per area.

## Scanning electron microscopic imaging

PC-3M cells and fibroblasts were plated on plastic coverslips (Sarstedt) and left to grow in normal media for 24 h. The media were then changed into experimental media as described before. The cells were fixed with 2.5% glutaraldehyde for 4 min and washed with Sörensen buffer (Morphisto). Immediately before HVSEM imaging, the coverslips were washed in aqua millipore and then exposed to

ascending series of ethanol (Merck) before critical point drying (Balzers CPD 030). The coverslips were mounted onto aluminium stubs using self-adhesive carbon discs (Baltic Praeperation). Coverslips were platinum sputter–coated (Bal-tec SCD 005; Pt-target: Baltic Praeperation) and introduced into the specimen chamber of the FEI Quanta 400 FEG electron microscope under high vacuum conditions. The angulation of the specimen to the beam was 90°. Images were taken using the Everhart-Thornley detector detector at a working distance of 10 mm and an accelerating voltage of 4 kV at 2,000× original magnification. Cells with round body and a distinct pseudopodium extended in one direction were considered as ameboidal, cells with no contact with other cells or cells with no more than 20% contact with others cells were considered as single.

## Lipidomics

Lipids were quantified by direct flow injection electrospray ionization tandem mass spectrometry (ESI-MS/MS) in positive ion mode using the analytical setup and strategy described previously (Liebisch et al, 2004, 2006). A fragment ion of m/z 184 was used for phosphatidylcholine (PC), sphingomyelin (SM) (Liebisch et al, 2004) and lysophosphatidylcholine (LPC) (Liebisch et al, 2002). The following neutral losses were applied: phosphatidylethanolamine (PE) 141, phosphatidylserine (PS) 185, phosphatidylglycerol (PG) 189, and phosphatidylinositol (PI) 277 (Matyash et al, 2008). PE-based plasmalogens (PE P) were analysed according to the principles described by Zemski-Berry and Murphy (2004). Sphingosine based ceramides (Cer) and hexosylceramides (HexCer) were analysed using a fragment ion of m/z 264 (Liebisch et al, 1999). Free cholesterol and cholesteryl ester were quantified using a fragment ion of m/z 369 after selective derivatization of free cholesterol (Liebisch et al, 2006). Lipid species were annotated according to the recently published proposal for shorthand notation of lipid structures that are derived from mass spectrometry (Liebisch et al, 2013). Glycerophospholipid species annotation was based on the assumption of even numbered carbon chains only. SM species annotation is based on the assumption that a sphingoid base with two hydroxyl groups is present.

## Statistical analysis for non-targeted metabolomics

Statistical analysis was performed using R (version 3.6.1) (R Core Team, 2018). Assay-specific quality control was used to correct for not available (NA) values, by minimum value imputation [min/sqrt(2)] with a random permutation. The randomizing factor was located in the range of 0.75 times to 1.25 times min/sqrt(2).

Metabolites which had to be excluded because of having more than 40% NA (entries for which the measurement did not yield any information as to the peak area) over all samples were 3,4-hydroxyphenyllactate, 4-acetamidobutanoate, creatinine, γ-glutamylisoleucine, γ-glutamylleucine, glutathione (oxidized) GSSG, hippurate, N-acetylmethionine, N1-methylguanosine, pro-hydroxy-pro, and pseudouridine as well as the unknown metabolites X-15382, X-15484, X-16266, X-17105, X-17121, and X-18225.

Statistics testing was carried out using Mann–Whitney U test with Bonferroni correction for multiple testing. Normal data distribution could not be detected by Shapiro–Wilk test. Significance was set at the $P < 0.05$ level, additionally a partial least squares-discriminant analysis (PLS-DA) was performed in R.

## Statistical analysis for targeted metabolomics

NA (entries for which the measurement did not yield any information as to the concentration) imputation was performed for metabolites with less than 40% missing values. Metabolites with more than 40% missing values were discarded. These excluded metabolites (or metabolite ratios) were found to be as follows:

ADMA, α-AAA, c4-OH-Pro, Carnosine, DOPA, Dopamine, Nitro-Tyr, SDMA, Serotonin, PC ae C42:4, SM (OH) C14:1, SM C26:0, ADMA/Arg, SDMA/Arg, and Serotonin/Trp.

Also, metabolites of the reference samples were checked for coefficients of variation of more than 25% in their respective reference samples. The following metabolites were detected to be in violation of this criterium and subsequently removed:

C5-DC (C6-OH), Histamine, PEA, Spermine, lysoPC a C26:0, lysoPC a C26:1, lysoPC a C28:0, lysoPC a C28:1, PC aa C24:0, PC aa C26:0, and PC ae C30:1.

Statistics testing was carried out using $t$ test with Benjamini–Hochberg correction for multiple testing. Significance was set at the $P < 0.05$ level. Log-normal data distribution was detected by Shapiro-Wilks normality test. In addition, a PLS-DA was performed in R.

### In vivo experiments

Mouse experiments were conducted according to the regulations of the State of Bavaria (permission granted by Regierung von Unterfranken, 55.2-2532.1-34/14 for subcutaneous [Mycielska et al, 2018] and AZ 55.2-2532-2-805 intrasplenic model, respectively). In the intrasplenic injection model, L3.6pl cells ($1 \times 10^5$ cells per animal) were injected into the lower spleen pole of athymic nude mice (13 animals per group; age: 6–8 wk, Crl:NU(NCr)-Foxn1nu, Charles River). To avoid side effects of intrasplenic "primary tumours," the spleen was removed 15 min after tumour cell injection. Treatment started on day 1 after tumour cell injection and comprised daily intraperitoneal injection of sodium gluconate (500 mg/kg/d); the control group was injected with NaCl only. After 24 d the mice were euthanized. Upon necropsy, livers were dissected and processed for further analyses. Hepatic tumour load was evaluated by a macroscopic score (0 = no tumour load; 1 = small singular tumours; 2 = large singular tumours; 3 = confluent tumours in less than half of liver; 4 = confluent tumours in more than half of liver; 5 = tumours in all liver segments).

The 3D in vivo tumour model (CAM, chorioallantoic membrane) was carried out as described in previous publications (Feder et al, 2020; Pion et al, 2020; Troebs et al, 2020). Fertilized chicken eggs were incubated in a ProCon egg incubator (Grumbach) at 37.8° C and 63% humidity under hourly rotation for a 4-d period until a window of ~1 × 1 cm was cut into the eggshell and sealed again with tape. After 4-d incubation, $2 \times 10^6$ human pancreas cancer cells (L3.6pl) suspended in 30 μl Matrigel (Corning) were grafted onto the CAM. After 2 d of tumour growth, tumour was daily treated with gluconate or NaCl (control) for 5 d before tumour excision. Tumour weight was determined and tumour volume was measured using Keyence VHX-7000 microscope (Keyence Germany, Neu-Isenburg, (Troebs et al, 2020)).

### Statistical analysis

Statistical analysis was performed using unpaired two-sided $t$ test unless otherwise stated. Details of statistical analysis for

non- and targeted metabolomics are included in the Supplemental Information.

## Supplementary Information

## Acknowledgements

We are very grateful to Prof. Paul McNeil (Augusta University, USA) for reviewing the manuscript and to Mr. Lindsay Parkinson for assistance with graphics. We would like to thank Silke Becker (Helmholtz, Munich), Bianca Eichner, Brigitte Wild (Regensburg Center for Interventional Immunology, Regensburg), Gerlinde Friestl and Helga Ebensberger (Department of Conservative Dentistry and Periodontology, Regensburg), and Vanessa Grage and Alexandra Werner (Institute of Pathology, Kaufbauren) for their excellent technical support. This work was supported by the Deutsche Forschungsgemeinschaft (DFG) grant (GE1188/5-1) to J Adamski and E Geissler and a DFG grant to S Haferkamp (Research group FOR 2127, Selection and adaptation during metastatic cancer progression, DFG), Else-Kröner-Fresenius-Stiftung grant to KM Schmidt and ReFoRmC to ME Mycielska. K Drexler is a fellow of the "Else-Kröner-Fresenius-Stiftung."

### Author Contributions

K Drexler: conceptualization, formal analysis, investigation, visualization, methodology, and writing—original draft, review, and editing.

KM Schmidt: conceptualization, formal analysis, funding acquisition, investigation, visualization, and methodology.

K Jordan: formal analysis, investigation, and visualization.

M Federlin: conceptualization, formal analysis, investigation, visualization, and methodology.

VM Milenkovic: conceptualization, formal analysis, investigation, visualization, and methodology.

G Liebisch: formal analysis, investigation, and methodology.

A Artati: formal analysis, investigation, and methodology.

C Schmidl: formal analysis, investigation, and methodology.

G Madej: conceptualization, formal analysis, investigation, visualization, and methodology.

J Tokarz: formal analysis, investigation, and methodology.

A Cecil: formal analysis, visualization, and methodology.

W Jagla: formal analysis and investigation.

S Haerteis: formal analysis, investigation, visualization, and methodology.

T Aung: formal analysis, investigation, and methodology.

C Wagner: formal analysis, investigation, and methodology.

M Kolodziejczyk: formal analysis, investigation, and methodology.

S Heinke: formal analysis, investigation, and methodology.

EH Stanton: investigation and methodology.

B Schwertner: investigation and methodology.

D Riegel: investigation.

CH Wetzel: resources and formal analysis.

W Buchalla: resources and methodology.

M Proescholdt: conceptualization and investigation.

CA Klein: conceptualization and writing—original draft.

M Berneburg: conceptualization, resources, funding acquisition, and writing—original draft.

HJ Schlitt: conceptualization, resources, funding acquisition, and writing—original draft.

T Brabletz: conceptualization and writing—original draft.

C Ziegler: conceptualization, resources, formal analysis, methodology, and writing—original draft.

EK Parkinson: conceptualization, formal analysis, and writing—original draft, review, and editing.

A Gaumann: conceptualization, resources, formal analysis, investigation, visualization, and writing—original draft, review, and editing.

EK Geissler: conceptualization, resources, funding acquisition, and writing—original draft.

J Adamski: conceptualization, resources, formal analysis, funding acquisition, investigation, methodology, and writing—original draft, review, and editing.

S Haferkamp: conceptualization, formal analysis, supervision, funding acquisition, investigation, methodology, and writing—original draft, review, and editing.

ME Mycielska: conceptualization, formal analysis, supervision, funding acquisition, investigation, methodology, and writing—original draft, review, and editing.

### Conflict of Interest Statement

ME Mycielska and EK Geissler are inventors on the Patent Application no. EP15767532.3 and US2020/408741 (status patent pending) and US2017/0241981 (patent issued) "The plasma membrane citrate transporter for use in the diagnosis and treatment of cancer" owned by the University Hospital Regensburg. No potential conflicts of interest are disclosed by the other authors.

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
