## [Reviewer comments · Life Science Alliance]

Life Science Alliance

Cancer-associated cells release citrate to support tumour metastatic progression

Konstantin Drexler, Katharina Schmidt, Katrin Jordan, Marianne Federlin, Vladimir Milenkovic, Gerhard Liebisch, Anna Artati, Christian Schmidl, M. Madej, Janina Tokarz, Alexander Cecil, Wolfgang Jagla, Silke Haerteis, Thiha Aung, Christine Wagner, Maria Kolodziejczyk, Stefanie Heinke, Evan Stanton, Barbara Schwertner, Dania Riegel, Christian Wetzel, Wolfgang Buchalla, Martin Proescholdt, Christoph Klein, Mark Berneburg, Hans Schlitt, Thomas Brabletz, Christine Ziegler, Eric Kenneth Parkinson, Andreas Gaumann, Edward Geissler, Jerzy Adamski, Sebastian Haferkamp, and Maria Mycielska

DOI: <https://doi.org/10.26508/lsa.202000903>

Corresponding author(s): Maria Mycielska, University Hospital Regensburg and Sebastian Haferkamp, University Hospital Regensburg

Review Timeline:

Submission Date:	2020-09-04
Editorial Decision:	2020-10-07
Appeal Received:	2021-01-15
Editorial Decision:	2021-02-04
Revision Received:	2021-02-05
Editorial Decision:	2021-03-01
Revision Received:	2021-03-09
Accepted:	2021-03-10

Scientific Editor: Shachi Bhatt

Transaction Report:

October 7, 2020

Re: Life Science Alliance manuscript #LSA-2020-00903-T

Dr. Maria Mycielska
University Hospital Regensburg
Surgery
Franz-Josef-Strauß-Allee 11
Regensburg 93053
Germany

Dear Dr. Mycielska,

Thank you for submitting your manuscript entitled "Cancer-associated cells release citrate to support tumour metastatic progression". The manuscript has been evaluated by expert reviewers, whose reports are appended below. Unfortunately, after an assessment of the reviewer feedback, our editorial decision is against publication in Life Science Alliance.

Although your manuscript is intriguing, we feel that the points raised by the reviewers are more substantial than can be addressed in a typical revision period. Given the interest in the topic, we would be open to resubmission to Life Science Alliance of a significantly revised and extended manuscript that fully addresses the reviewers' concerns and is subject to further peer-review. If you would like to resubmit this work to Life Science Alliance, please submit the revised manuscript as an appeal directly through our manuscript submission system. Please note that priority and novelty would be reassessed at resubmission.

Regardless of how you choose to proceed, we hope that the comments below will prove constructive as your work progresses. We would be happy to discuss the reviewer comments further once you've had a chance to consider the points raised in this letter.

Thank you for thinking of Life Science Alliance as an appropriate place to publish your work.

Sincerely,

Shachi Bhatt, Ph.D.
Executive Editor
Life Science Alliance
<https://www.lsajournal.org/>
Tweet @SciBhatt @LSAJournal

Reviewer #1 (Comments to the Authors (Required)):

This manuscript employs a range of approaches to investigate how citrate might be involved in tumor-stroma crosstalk and metastasis. The question is interesting, but the manuscript is a rather mixed bag with some parts convincing and other parts very weak. In brief, the authors argue that

extracellular citrate influences cancer cells in a variety of ways and a plasma membrane citrate transporter is expressed in some stromal cells. This leads to the argument that stromal cells produce citrate that then affects cancer cells. This work is a continuation of previous studies, most notably the authors 2018 Cancer Research paper. The main weaknesses of the study are an over-reliance on gluconate to inhibit the citrate channel and the over-stating of equivocal data.

Main comments

1. Figure 1B top panel does not show a convincing difference in pmCiC + or - citrate.
2. It would be helpful to know the relative levels of cytokines produced by the cancer cells and fibroblasts. It is not clear if the pixel density measurement allows for comparison between the two plots.
3. Figure 2A shows no consistent difference between the + and - citrate samples. Also, the legend is not sufficient to fully understand what is being shown. Is each row a replicate?
4. Figure 3A&D do not show convincing differences in EMT regulators. The authors should provide quantification for multiple (three or more) biological replicates. Furthermore, the biological significance of such small changes is hard to gauge.
5. The differences in cell morphology in the SEM images in Figure 3B are not clear. The authors should provide some morphological quantification to back up their claims.
6. If performed properly, then intrasplenic injection of cancer cells followed by splenectomy should not result in the growth of tumors at the wound site. The fact that 60% of control mice develop tumors at their wounds is concerning.
7. Gluconate is a pleiotropic molecule and cannot be considered a specific perturbation. I realize that it might not be totally straightforward, but the authors need to devise and use a more specific way of targeting pmCiC. Can Crispr or RNAi be used against the exon that results in plasma membrane targeting? Thereby leaving the mitochondrial isoform unaffected.
8. Why discuss the ATAC seq data if it is inconclusive and peripheral to the main message?
9. The data in Figure 6 are anecdotal and not really convincing.
10. The model in Supp. Figure 11 is confusing and seems to lack the colors referred to in the legend.

Reviewer #2 (Comments to the Authors (Required)):

In the study by Drexler et al. the authors report that citrate can be transferred from CAF to cancer cells through the pmCiC transporter. This mechanism is activated when cancer cells are cultured in the absence of citrate, which triggers cancer cell production of several cytokines including TGF-beta to alter CAF, leading to enhanced secretion of CAF-derived citrate and cytokines. This is believed to stimulate an invasive phenotype in cancer cells and promote tumor growth and metastasis in vivo. It is shown that treatment with gluconate, a specific inhibitor of pmCiC, suppresses the growth and metastasis of human pancreatic cancer cells in a mouse model. Overall,

this work is original and the findings are interesting. A lot of data are included but they don't seem to be cohesive and are weak in supporting the specific role of citrate, and particularly CAF-derived citrate, in supporting tumor progression. Some experimental settings are not fully supported for a physiological relevance. I feel that the following concerns would need to be addressed before acceptance by LSA for publication.

Major points:

1. Throughout the manuscript, the authors examined the effect of citrate deprivation to show that this metabolic stress in cancer cells alters CAF metabolism and citrate export through paracrine signaling. However, deprivation of citrate can hardly occur by itself without simultaneous deprivation of other nutrients/metabolites in the tumor microenvironment. In addition, under in vivo conditions, as cancer cells and stroma cells share the same metabolic environment, when cancer cells are deprived of citrate (and other nutrients), CAF would be under the same metabolic stress. Therefore, the "- citrate" treatment cannot reflect a real physiological condition. Other conditions such as deprivation of glucose and amino acids should be added, and for citrate deprivation other metabolites such as pyruvate and other TCA cycle intermediates should be included as controls. Cancer cells and CAF should be tested under the same metabolic condition for CM transfer experiment and others.

2. Intracellular citrate can be quickly restored through TCA cycle. It is difficult to believe that lack of exogenous citrate (deprivation) and transfer of CAF-derived citrate would have a potent effect. For example, even cancer cells are grown in citrate-free medium, they should be able to immediately generate citrate, making the +/- citrate treatment less meaningful.

3. It is unclear if the concentration of exogenous citrate used in treatment is comparable to endogenous level in the extracellular space. It is also unclear to what extent extracellular citrate levels fluctuate in the tumor microenvironment.

4. In addition to citrate, many other metabolites are increased in the CM of CAF following cancer cell reprogramming. There is not sufficient evidence supporting that citrate is the major contribution from CAF to cancer cells. Many other metabolites can be transferred from CAFs to cancer cells, as well-demonstrated by others.

5. It is unclear if citrate transferred from CAF to cancer cells eventually enters fatty acid synthesis, energy production, or epigenetic regulation. Using isotope labeled citrate may help address this question.

6. Mouse tumor experiment with gluconate treatment does not necessarily support the proposed model, as the effect of drug treatment is systemic, and there is no way to determine the flux of citrate between different types of cells. Other mouse tumor models should be included and the citrate exchange between CAF and cancer cells needed to be determined.

Minor points:

1. The size of text in figures should be more consistent.

Reviewer #3 (Comments to the Authors (Required)):

The manuscript by Drexler et al. claims that cancer associated fibroblast releases citrate to support tumor metastatic progression. They claim that the main metabolic role of CAF is to synthesize and

release citrate. While the authors show extensive amount of in vitro and in vivo data, the conclusions are not well supported by the data. It is clear that the citrate transporter inhibitor gluconate is able to impede cancer growth, but the data presented are not sufficient to demonstrate that the main metabolic role of CAF is to synthesize and release citrate. The manuscript is not very well written, it is difficult for this reviewer to follow the manuscript. Many data are confusing.

Major concerns:

1. Fig. 1A is confusing. The authors claim that "the amount of citrate released and in consequence CAFs metabolic activity is determined by the availability of citrate to cancer cells". It is not clear to this reviewer why preconditioning cancer cells with or without citrate affects the amount of citrate released from CAFs. No data were shown to directly demonstrate the amount of citrate released by CAFs.
2. Fig. 3 shows that citrate transiently and modestly increases EMT phenotype. The functional significance of citrate released by CAF is mild to this reviewer.
3. In Fig. 4A, the hepatic tumour load difference is statistically significant but modest.
4. In Fig. 4C. aSMA staining alone is not sufficient to claim stromal transformation. Does gluconate have direct impact on aSMA expression?
5. The authors claim that citrate is important to support tumor metastatic progression, but the metastasis data are not convincing.

Dear Dr. Bhatt,

The authors of manuscript #LSA-2020-00903-T have requested an appeal. Their comments are below.

Dear Dr. Bhatt,

Thank you very much for your decision letter which we have received on the 7.10.20 and giving us the opportunity to re-submit revised version of the manuscript entitled "Cancer-associated cells release citrate to support tumour metastatic progression" in the form of an appeal.

We thank the reviewers for their comments. We have carefully read and considered all the comments of all the three Reviewers and made the requested changes to the manuscript. Altogether, we believe that a more analytical presentation of our data together with additional experiments make our work more convincing and more complete. As you will see from the detailed letter as below we were able to perform most of the requested experiments including an additional animal model for which we used CAM assay. Unfortunately, due to the as yet unknown regulation of pmCiC expression we are unable to specifically silence the plasma membrane citrate transporter without decreasing the expression of its mitochondrial counterpart. We have used all the available options including transient and stable siRNA transfections as well as CRISPR-Cas9 but unfortunately, specific silencing was not possible (all the data are available upon request). Based on these data we do assume that the first steps of the transcription of these two genes must be common. However, as we also explain below we do not agree with the referee's assertion that gluconate is an unspecific blocker of the pmCiC in cancer cells and therefore, can be used to study the effects of citrate deprivation on cancer cells.

The following main changes were made to the manuscript:

- (1) More analytical presentation of the data.
- (2) Measurements of the levels of citrate released by fibroblasts stimulated by cancer cells preincubated with or without extracellular citrate were made.
- (3) To show more specifically the effects of blocking of citrate uptake on cancer cells in vivo we have performed additional staining using fibroblasts activating protein (FAP) of metastasis. We are now able to show not only decreased stroma transformation in gluconate-treated animals but also an increased immune response, consistent with the observed increased release of the pro-inflammatory cytokines by cancer cells deprived of extracellular citrate as measured in vitro.
- (4) Additional animal model. In the German system it takes over a year to obtain a new ethics permission but more importantly, the studied protein, pmCiC is human- and higher primate-specific and for that reason we do not believe that additional mouse experiments will shed more light on the action of extracellular citrate in cancer progression. We also believe that for this very reason the potential anti-cancer effects in humans might be even more pronounced than the results seen in animals (injected with human cancer cells). However, we have performed a CAM assay to address the

Reviewer's request.

We hope you will find this manuscript much improved and look forward to hearing from you,

Kind regards,
Maria Mycielska and Sebastian Haferkamp

MS: LSA-2020-00903-T

Dr. Maria Mycielska
University Hospital Regensburg
Surgery
Franz-Josef-Strauß-Allee 11
Regensburg 93053
Germany

Dear Dr. Mycielska,

Thank you for submitting an appeal for your manuscript entitled "Cancer-associated cells release citrate to support tumour metastatic progression" that was previously peer-reviewed at Life Science Alliance (LSA).

We have now evaluated your appeal letter, revisions and point-by-point response, and we think that the manuscript has been improved sufficiently to send back to the original referees for a re-evaluation. Of course, the final decision for publication will depend on whether the reviewers' evaluation.

We now ask you submit the revised version of the manuscript using the link below that we can then send back to the referees.

Please use the following link to submit your revised manuscript:
<https://lsa.msubmit.net/cgi-bin/main.plex?el=A5Na1XU7A7CS1c5l6B9ftdg3fuVGIBn1xP0Go4OoEAZ>

Yours sincerely,

Shachi Bhatt, Ph.D.
Executive Editor
Life Science Alliance
<https://www.lsjournal.org/>
Tweet @SciBhatt @LSAjournal

Interested in an editorial career? EMBO Solutions is hiring a Scientific Editor to join the international Life Science Alliance team. Find out more here -
https://www.embo.org/documents/jobs/Vacancy_Notice_Scientific_editor_LSA.pdf

Dear Dr. Bhatt,

Thank you very much for accepting our appeal and assessing our revised manuscript "Cancer-associated cells release citrate to support tumour metastatic progression" as sufficiently improved for re-evaluation.

We thank the reviewers for their comments. We have carefully read and considered all the comments of all the three Reviewers and made the requested changes to the manuscript. Altogether, we believe that a more analytical presentation of our data together with additional experiments make our work more convincing and more complete. As you will see from the detailed letter as below we were able to perform most of the requested experiments including an additional animal model for which we used CAM assay. Unfortunately, due to the as yet unknown regulation of pmCiC expression we are unable to specifically silence the plasma membrane citrate transporter without decreasing the expression of its mitochondrial counterpart. We have used all the available options including transient and stable siRNA transfections as well as CRISPR-Cas9 but unfortunately, specific silencing was not possible (all the data are available upon request). Based on these data we do assume that the first steps of the transcription of these two genes must be common. However, as we also explain below we do not agree with the referee's assertion that gluconate is an unspecific blocker of the pmCiC in cancer cells and therefore, can be used to study the effects of citrate deprivation on cancer cells.

The following main changes were made to the manuscript:

- (1) More analytical presentation of the data.
- (2) Measurements of the levels of citrate released by fibroblasts stimulated by cancer cells preincubated with or without extracellular citrate.
- (3) To show more specifically the effects of blocking of citrate uptake on cancer cells *in vivo* we have performed additional staining using fibroblasts activating protein (FAP) of metastasis. We are now able to show not only decreased stroma transformation in gluconate-treated animals but also an increased immune response, consistent with the observed increased release of the pro-inflammatory cytokines by cancer cells deprived of extracellular citrate as measured *in vitro*.
- (4) Additional animal model. In the German system it takes over a year to obtain a new ethics permission but more importantly, the studied protein, pmCiC is human- and higher primate-specific and for that reason we do not believe that additional mouse experiments will shed more light on the action of extracellular citrate in cancer progression. We also believe that for this very reason the potential anti-cancer effects in humans might be even more pronounced than the results seen in animals (injected with human cancer cells). However, we have performed a CAM assay to address the Reviewer's request.

Reviewer #1 (Comments to the Authors (Required)):

This manuscript employs a range of approaches to investigate how citrate might be involved in tumor-stroma crosstalk and metastasis. The question is interesting, but the manuscript is a rather mixed bag with some parts convincing and other parts very weak. In brief, the authors argue that extracellular citrate influences cancer cells in a variety of ways and a plasma membrane citrate

transporter is expressed in some stromal cells. This leads to the argument that stromal cells produce citrate that then affects cancer cells. This work is a continuation of previous studies, most notably the authors 2018 Cancer Research paper. The main weaknesses of the study are an over-reliance on gluconate to inhibit the citrate channel and the over-stating of equivocal data.

Main comments

1. Figure 1B top panel does not show a convincing difference in pmCiC + or - citrate.

Answer: We now show the absolute values of citrate released by fibroblasts into the media – Fig. 1B, p. 6-7.

2. It would be helpful to know the relative levels of cytokines produced by the cancer cells and fibroblasts. It is not clear if the pixel density measurement allows for comparison between the two plots.

Answer: The assay was used according to the manufacturer instructions and analysed using supplied positive and negative controls for all the assays. We used the same control media for all conditions tested, therefore we believe it is possible to use the values shown in the figures for the comparative analysis of the relative values of cytokines in cancer cells and fibroblasts, p. 21

3. Figure 2A shows no consistent difference between the + and - citrate samples. Also, the legend is not sufficient to fully understand what is being shown. Is each row a replicate?

Answer: We have re-analysed the data in detail and concentrated on citrate versus control groups and show the statistics for all the remaining groups (Fig. 2A; Supp. Fig. 6&7). Thanks to that we have determined that incubation of prostate cancer cells with citrate reduces intracellular levels of the metabolites known to be used for intracellular citrate synthesis (such as glutamine, glutamate, serine or glycine) which is in line with our hypothesis that part of the extracellular citrate is supplied to cancer cells from outside, p9. We are grateful for this comment.

4. Figure 3A&D do not show convincing differences in EMT regulators. The authors should provide quantification for multiple (three or more) biological replicates. Furthermore, the biological significance of such small changes is hard to gauge.

Answer: We have now performed the analysis as suggested by the Reviewer and included quantifications of the biological replicates not only for this part of the study but also for all the remaining parts where western blot analysis was used (Fig. 3A&D, Supp. Fig. 11). We do see consistent changes of the pattern of metabolic characteristics and EMT marker expression. Although, we do agree that not all the markers show significant changes but together with the metabolomics/metabolic and morphological data they stay in line with the observation that incubation with extracellular citrate stimulates acquisition of a more aggressive character of the cells as compared to the control.

5. The differences in cell morphology in the SEM images in Figure 3B are not clear. The authors should provide some morphological quantification to back up their claims.

Answer: We have now analysed the pictures according to the papers dealing with the issue of invasion and colonisation and the data are presented in Fig. 3B and also in Supp. Fig. 10.

6. If performed properly, then intrasplenic injection of cancer cells followed by splenectomy should not result in the growth of tumors at the wound site. The fact that 60% of control mice develop tumors at their wounds is concerning.

Answer: L3.6pl cells are a very aggressive cancer cell type and some wound tumours are known to occur. We now discuss this result and show the literature supporting our statement, p. 13.

7. Gluconate is a pleiotropic molecule and cannot be considered a specific perturbation. I realize that it might not be totally straightforward, but the authors need to devise and use a more specific way of targeting pmCiC. Can Crispr or RNAi be used against the exon that results in plasma membrane targeting? Thereby leaving the mitochondrial isoform unaffected.

Answer: This is of course a very important question and for several years we have struggled to achieve specific silencing of the pmCiC. We have tried the following:

1. siRNA. We had siRNA designed by Eurofins and used all the options they suggested, however, due to a relatively small sequence difference between the two proteins and exceptionally high content of Cs and Gs none of the possibilities offered were optimal. Nonetheless, we used transient and permanent transfections, unfortunately both proteins pmCiC and mCiC expression has been affected. For this reason, we have used transient siRNA transfection for the citrate transport studies only as in this case decrease of mCiC would have no influence on the results (Mazurek et al., 2010; Mycielska et al., 2018).

2. We hoped that using transient transfections and adjusting the amount of the siRNA we could control more the effects on pmCiC versus mCiC. We performed additional studies to choose the most promising concentrations of the siRNA and performed animal studies using AteloGene (Koken) kit for transient transfections. Unfortunately, the reduction in mCiC was still prominent, which would make the obtained results unspecific.

3. CRISPR. We have chosen the first exon of the pmCiC to check several possibilities of CRISPR-Cas9. We have screened many clones and in all cases, reduction in pmCiC was accompanied by reduced mCiC expression. Importantly we have not found any clones with no expression of pmCiC suggesting that complete removal is lethal to the cells consistent with an interdependent regulation of these two genes.

To conclude, we are still not sure how pmCiC/mCiC expression is specifically regulated. Our results suggest that at the first steps, the two genes might be transcribed together and additional regulation is employed which allows for selecting one or the other for further processing. Unfortunately, this is a complex issue and out of the scope of the present study.

Gluconate specificity

Answer: Gluconate is plasma membrane impermeable, has been accepted as a physiologically neutral substance (by FDA), and is used widely in biology and medicine. In medicine gluconate is used as a carrier of metals such as Zn²⁺, Mg²⁺ or Ca²⁺ even at very high concentrations (infusions of Ca²⁺ gluconate contain 1 g of gluconate). In electrophysiology and transport experiments gluconate is routinely used to replace chloride because of its lack of effects on the cells. Moreover, our current experiments depicted in the manuscript determined that gluconate affects citrate transport in cancer cells only, even though pmCiC expression occurs also in prostate epithelial cells and CAFs but with a

function of citrate export. Indeed, when applied alone (without citrate) at very high and totally unphysiological concentrations gluconate seems to have an effect on cancer cell mitochondria possibly through forced entrance through pmCiC (gluconate has a similar structure to citrate, therefore at high concentrations and in the absence of citrate some of it might be transported through pmCiC).

In the present ms we show also that pmCiC has two binding sites for gluconate which are not the same as the citrate binding site (Fig. 2C). One of them is located in its N-terminus which is hanging freely, most likely in the extracellular space. This N-terminus is long and unusually heavily charged suggesting that it has a role either in regulating citrate transport, trafficking or also potentially some signalling/receptor functions. Binding of gluconate will affect the structure of N-terminus which might be followed by some additional effects. However, it still means that the only affected protein by gluconate is pmCiC and only in cancer cells, therefore, making gluconate a rather specific inhibitor. We are in the process of studying pmCiC structure and its functions further as well as preparing for a drug screen in cancer cells and CAFs, however, this issue is outside the scope of this manuscript.

8. Why discuss the ATAC seq data if it is inconclusive and peripheral to the main message?

Answer: These data have now been removed.

9. The data in Figure 6 are anecdotal and not really convincing.

Answer: Fig. 6 includes 6 examples of pmCiC immunohistochemical staining of pancreatic and gastric cancers. These cancers are known to have strong desmoplastic fibrosis with abundant cancer associated fibroblasts (CAFs) and accordingly we see the pmCiC positivity in numerous CAFs in these tissues. For the purpose of this study we analysed 25 gastric carcinomas (diffuse and intestinal type) as well as 25 pancreatic ductal adenocarcinomas. As shown in figure 6 we could see the most prominent pmCiC expression in cancer cells at the invasion front but also in the spindle shaped cells of the stromal reaction, which are mainly CAFs. In addition, we have stained 32 samples of head and neck squamous cell cancer (HSNCC), 59 invasive/non-invasive bladder cancer, 156 metastatic and non-metastatic breast cancer (hormone receptor positive/negative and HER-2 types), 39 bone and lymph node metastasis of breast and prostate cancer, 6 Glioblastomas, 56 metastatic and non – metastatic colon cancer tissues and found an expression of pmCiC in cancer cells and also in cells of the tumor microenvironment (TME). We are preparing an in depth analysis of the tissues regarding pmCiC expression in cancer cells and in the TME for a separate manuscript as we believe this is outside the scope of the present project.

We have now added higher magnifications of the immunohistochemical staining to Fig. 6 to better highlight the expression of pmCiC in the TME. Although we agree with the reviewer that more staining is needed to better understand the role of pmCiC in CAF 's in human cancer, we are convinced that these data together with the *in vitro* and animal experiments show that pmCiC is present in CAF's and plays a role in the complex interplay of cancer cells with the TME.

10. The model is Supp. Figure 11 is confusing and seems to lack the colours referred to in the legend.

Answer: Thank you for spotting this mistake. We have amended the diagram to make it easier and added the colours.

Reviewer #2 (Comments to the Authors (Required)):

In the study by Drexler et al. the authors report that citrate can be transferred from CAF to cancer cells through the pmCiC transporter. This mechanism is activated when cancer cells are cultured in the absence of citrate, which triggers cancer cell production of several cytokines including TGF-beta to alter CAF, leading to enhanced secretion of CAF-derived citrate and cytokines. This is believed to stimulate an invasive phenotype in cancer cells and promote tumor growth and metastasis in vivo. It is shown that treatment with gluconate, a specific inhibitor of pmCiC, suppresses the growth and metastasis of human pancreatic cancer cells in a mouse model. Overall, this work is original and the findings are interesting. A lot of data are included but they don't seem to be cohesive and are weak in supporting the specific role of citrate, and particularly CAF-derived citrate, in supporting tumor progression. Some experimental settings are not fully supported for a physiological relevance. I feel that the following concerns would need to be addressed before acceptance by LSA for publication.

Major points:

1. Throughout the manuscript, the authors examined the effect of citrate deprivation to show that this metabolic stress in cancer cells alters CAF metabolism and citrate export through paracrine signaling. However, deprivation of citrate can hardly occur by itself without simultaneous deprivation of other nutrients/metabolites in the tumor microenvironment.

Answer: We believe there might be a misunderstanding due to an unclear description provided by us in the first version of the manuscript and we now added additional explanation. At no point of the applied experimental procedures have we deprived the cells of intracellular citrate, we have only removed extracellular citrate leaving the cells all the necessary metabolites to produce citrate intracellularly. Moreover, to be absolutely sure and unlike other researchers at every step of the experimental procedure we were adding additional glucose and glutamine (we have no updated Supp. Fig. 3 to make the procedures more clear), therefore the metabolites known to be necessary to produce citrate intracellularly. We do agree with the Reviewer that intracellular citrate deprivation would be totally unphysiological, however, depriving cancer cells or CAFs of citrate in general was not the purpose of the study and such conditions have never been applied. It is now explained better in the manuscript, p. 6).

In addition, under in vivo conditions, as cancer cells and stroma cells share the same metabolic environment, when cancer cells are deprived of citrate (and other nutrients), CAF would be under the same metabolic stress. Therefore, the "- citrate" treatment cannot reflect a real physiological condition. Other conditions such as deprivation of glucose and amino acids should be added, and for citrate deprivation other metabolites such as pyruvate and other TCA cycle intermediates should be included as controls. Cancer cells and CAF should be tested under the same metabolic condition for CM transfer experiment and others.

Answer: Although we agree with the Reviewer in general that cancer cells and cancer-associated stroma are subject to similar extracellular metabolic conditions, we cannot agree that this would induce the same metabolic stress in the two cell types. While cancer cells need extracellular metabolites to support their increased needs for nutrients and energy consumption, cancer-associated stroma is used to support cancer cells, therefore to synthesise the necessary metabolites or obtain them in the process of mitophagy/autophagy. The two cell types are therefore unlikely to respond to the changes in the extracellular metabolite levels in the same way. Our study shows that extracellular citrate deprivation affects cancer cells metabolism, however, we could not find any evidence in our data or in the literature that absence of extracellular citrate would in any way affect CAFs.

As for controls, we believe it is an important question and we now present an additional analysis of our metabolomic data showing that indeed, there is an increase of serine release by cancer cells grown without extracellular citrate consistent with the increased intracellular citrate synthesis. Accordingly, intracellular levels of glutamate, glutamine, serine and glycine are increased in cancer cells deprived of extracellular citrate. We did not observe any changes in the level of extracellular pyruvate when comparing the media obtained from cancer cells with the media from CAFs suggesting that pyruvate does not play any significant role in the exchange between the two cell types. This additional analysis is now included in the manuscript and discussed, p. 6 (Fig. 2A and Supp. Fig. 1 and 6).

We have also performed measurements of the citrate released by CAFs and determined that they produce between 200-300 μM of this metabolite. To our best of knowledge this is the first measurement of citrate in the cancer environment and based on this data we assume that 200 μM of citrate used by us should be considered physiological. Moreover, this is also the level found in blood.

2. Intracellular citrate can be quickly restored through TCA cycle. It is difficult to believe that lack of exogenous citrate (deprivation) and transfer of CAF-derived citrate would have a potent effect. For example, even cancer cells are grown in citrate-free medium, they should be able to immediately generate citrate, making the +/- citrate treatment less meaningful.

Answer: Yes, we fully agree with this statement. Cancer cells are able to overcome the lack of extracellular citrate. The conditions used for these experiments had the purpose of securing unrestricted access of metabolites to enable unlimited intracellular citrate synthesis by cancer cells. Although, cancer cells are able to produce citrate intracellularly due to their metabolic flexibility this citrate synthesis requires significant changes in their metabolism (Mycielska et al., 2018; Haferkamp et al., 2020). Consistently our metabolomic data show increased levels of substrates involved in citrate synthesis when extracellular citrate is absent, p. 9, Fig 2A and Supp. Fig. 6.

3. It is unclear if the concentration of exogenous citrate used in treatment is comparable to endogenous level in the extracellular space. It is also unclear to what extent extracellular citrate levels fluctuate in the tumor microenvironment.

Answer: We have now measured the amount of citrate released by CAFs and evaluated it to be between 200-300 μM which stays in line with the amount used by us for the purpose of the experiments (Fig. 1B).

4. In addition to citrate, many other metabolites are increased in the CM of CAF following cancer cell reprogramming. There is not sufficient evidence supporting that citrate is the major contribution from CAF to cancer cells. Many other metabolites can be transferred from CAFs to cancer cells, as well-demonstrated by others.

Answer: Again, we fully agree with the statement that other metabolites are also transferred to cancer cells or at least present in the media obtained from CAFs e.g. amino acids pp. 5-6. We show and discuss these data in relation to the published reports (supplementary Figure 1). However, our data show also that CAFs metabolic activity depends on whether cancer cells had access to extracellular citrate before conditioning in the presence of all the other metabolites allowing for normal metabolic functioning including citrate synthesis (among them also pyruvate and glucose). We have softened the statement throughout the manuscript.

5. *It is unclear if citrate transferred from CAF to cancer cells eventually enters fatty acid synthesis, energy production, or epigenetic regulation. Using isotope labeled citrate may help address this question.*

Answer: We have already performed these experiments and published in our previous paper in Cancer Research (Mycielska et al., 2018). Indeed, we have determined that the uptake of extracellular citrate increases protein synthesis as well as modifies Krebs cycle activity. In the present manuscript, we show also that cancer cells incubated with extracellular citrate produce more of the plasma membrane related fatty acids (Fig. 3C).

6. *Mouse tumor experiment with gluconate treatment does not necessarily support the proposed model, as the effect of drug treatment is systemic, and there is no way to determine the flux of citrate between different types of cells. Other mouse tumor models should be included and the citrate exchange between CAF and cancer cells needed to be determined.*

Answer: We have shown that gluconate has a specific effect on cancer cells pmCiC only and it does not affect citrate transport in any other cell types studied. Accordingly, mitotoxicity studies performed on fibroblasts and CAFs included in the present manuscript show that gluconate has no effect on other cell types. Therefore, the treatment we performed is not systemic but blocks citrate uptake by cancer cells exclusively. In the German system it takes around a year to obtain an ethics permission for animal studies which limits our ability to use another mouse model. To show the effects on cancer growth in another animal model we have performed a CAM assay which is used as an alternative to mouse studies. Our results show a highly significant decrease of the tumour growth in the presence of gluconate (Fig. 5E). We hope that the animal data taken together are now more convincing.

Minor points:

1. The size of text in figures should be more consistent.

Answer: Thank you for this comment. We have now unified as much as we could the text in the figures.

Reviewer #3 (Comments to the Authors (Required)):

The manuscript by Drexler et al. claims that cancer associated fibroblast releases citrate to support tumor metastatic progression. They claim that the main metabolic role of CAF is to synthesize and release citrate. While the authors show extensive amount of in vitro and in vivo data, the conclusions are not well supported by the data. It is clear that the citrate transporter inhibitor gluconate is able to impede cancer growth, but the data presented are not sufficient to demonstrate that the main metabolic role of CAF is to synthesize and release citrate. The manuscript is not very well written, it is difficult for this reviewer to follow the manuscript. Many data are confusing.

We have made some major changes to the manuscript including more analytical analysis of the data, additional animal experiments, measurements of the absolute values of citrate released by CAFs. We

have also substantially rewritten the manuscript and modified the figures and the figure legends and hope that you will agree that this version of the manuscript is improved and easier to follow.

Major concerns:

1. Fig. 1A is confusing. The authors claim that "the amount of citrate released and in consequence CAFs metabolic activity is determined by the availability of citrate to cancer cells". It is not clear to this reviewer why preconditioning cancer cells with or without citrate affects the amount of citrate released from CAFs. No data were shown to directly demonstrate the amount of citrate released by CAFs.

Answer: We have shown previously (Mycielska et al., 2018) that cancer cells have the ability to take up extracellular citrate to support their metabolism, therefore extracellular citrate imposes significant changes to the overall status of cancer cells. In particular, we have found that extracellular citrate uptake decreases ROS synthesis as well as supports protein and fatty acid synthesis *in vivo*. In the present study we aimed to evaluate the consequences of changes in cancer cell metabolism induced by the uptake of extracellular citrate on their interactions with the environment. Our data show that cancer cells deprived of extracellular citrate employ different ways to communicate their need of citrate to their environment while cancer cells with extracellular citrate supply undergo metastatic changes. We have also performed citrate measurements in the media from fibroblasts stimulated by cancer cells pretreated with or without extracellular citrate and show, the exact values (p. 9, Fig. 1B).

2. Fig. 3 shows that citrate transiently and modestly increases EMT phenotype. The functional significance of citrate released by CAF is mild to this reviewer.

Answer: We now show the changes in the expression of the markers of the EMT as well as analysis of the SEM pictures in an analytical way (Figs 3A and B). We believe that this analysis is better suited to show that although not all the markers are affected by citrate pre-treatment the changes induced by extracellular citrate are consistent with the acquisition of a more aggressive character of cancer cells. It has to be taken into account that for the present study we used the most aggressive prostate cancer cell line which is likely to have some pathways/markers already highly active. Moreover, these data should be considered together with the metabolic and phenotypic changes which are all consistent with a more invasive status of the cells in the presence of extracellular citrate.

3. In Fig. 4A, the hepatic tumour load difference is statistically significant but modest.

Answer: We believe that similarly as our answer to the previous point, changes in the tumour load should be considered together with other characteristics of the subcutaneous and metastatic growth to fully assess the effects of decreased citrate uptake by cancer cells *in vivo*. Similarly, as in the case of the subcutaneous growth, the observed differences related not only to the size and to number of metastasis but also to their structure and characteristics. As suggested by the Reviewer in the following point, we have additionally used anti FAP antibody to study in more detail stroma transformation of liver metastasis in the two experimental groups (Fig. 4D and E). Our results show that there is a prominent stroma transformation in the case of control liver metastasis but this is not the case in the gluconate group. Moreover, we have observed an increased immune cells infiltration in the liver metastasis of the gluconate group. Therefore, it can be assumed that although the decrease in the number of liver metastasis in the gluconate treated group might be considered as

modest it should be assessed together with other characteristics of the growths such as stroma transformation, angiogenesis and immune infiltration. Moreover, to strengthen our data we have also performed a CAM assay and show highly significant decrease of the growth rate of tumours treated with gluconate (Fig. 5D).

4. In Fig. 4C. α SMA staining alone is not sufficient to claim stromal transformation. Does gluconate have direct impact on α SMA expression?

Answer: As above. We have checked the expression of the α SMA in fibroblasts grown in the presence of citrate or gluconate and no difference was observed (Supp. Fig. 11B).

5. The authors claim that citrate is important to support tumor metastatic progression, but the metastasis data are not convincing.

Answer: As explained above we believe that to correctly evaluate changes in the metastatic spread of pancreatic cancer cells *in vivo* not only the growth rate should be taken into account but also characteristics of metastasis. We have provided additional evidence (Figs. 4 and 5) together with a CAM assay to support our data.

March 1, 2021

RE: Life Science Alliance Manuscript #LSA-2020-00903-TR-A

Dr. Maria Mycielska
University Hospital Regensburg
Surgery
Franz-Josef-Strauß-Allee 11
Regensburg 93053
Germany

Dear Dr. Mycielska,

Thank you for submitting your revised manuscript entitled "Cancer-associated cells release citrate to support tumour metastatic progression". We would be happy to publish your paper in Life Science Alliance pending final revisions necessary to meet the remaining concerns laid out by Reviewers 1 and 2 and our formatting guidelines.

Along with the points listed below, please also attend to the following,

- please consult our manuscript preparation guidelines <https://www.life-science-alliance.org/manuscript-prep> and make sure your manuscript sections are in the correct order
- please add ORCID ID for the corresponding (and secondary corresponding) author--you should have received instructions on how to do so
- please add an Author Contributions section to your main manuscript text
- please revise the legend for figure S3; S12 so that the panels are introduced in order
- please add callouts for Figures 6A-F; S3A; S8A-G; S9A, B; S12A,B to your main manuscript text
- please add your supplementary figure legends to the main manuscript text after the main figure legends
- please provide higher resolution images for the blots shown in Figure 1C, 3A, S11A and S11B
- please improve the presentation of scale bars in Figure 4D so that they are more visible
- please provide the table shown in Figure S7B in an editable format

A. FINAL FILES:

B. MANUSCRIPT ORGANIZATION AND FORMATTING:

Sincerely,

Shachi Bhatt, Ph.D.
Executive Editor

Life Science Alliance

<https://www.lsa-journal.org/>

Interested in an editorial career? EMBO Solutions is hiring a Scientific Editor to join the international Life Science Alliance team. Find out more here -

https://www.embo.org/documents/jobs/Vacancy_Notice_Scientific_editor_LSA.pdf

Reviewer #1 (Comments to the Authors (Required)):

Overall, this work is improved and there is enough in it to warrant publication. However, it remains let down by confusing presentation and there are numerous fairly simple things that should be done to improve clarity and rigor. I realize that this is a bit long winded, but I have included both my original comment, the response, and my new response below.

1. Figure 1B top panel does not show a convincing difference in pmCiC + or - citrate.

Answer: We now show the absolute values of citrate released by fibroblasts into the media - Fig. 1B, p. 6-7.

OK

2. It would be helpful to know the relative levels of cytokines produced by the cancer cells and fibroblasts. It is not clear if the pixel density measurement allows for comparison between the two plots.

Answer: The assay was used according to the manufacturer instructions and analysed using supplied positive and negative controls for all the assays. We used the same control media for all conditions tested, therefore we believe it is possible to use the values shown in the figures for the comparative analysis of the relative values of cytokines in cancer cells and fibroblasts, p. 21.

The authors seem to have only partially answered the question. It is also critical that the exposure times are equivalent and the analyses performed in parallel. Having said that, the analysis presented is OK so long as the authors only make claims for each cell type alone +/- citrate and do not try to claim absolute differences in the levels between cancer cells and CAFs.

3. Figure 2A shows no consistent difference between the + and - citrate samples. Also, the legend is not sufficient to fully understand what is being shown. Is each row a replicate? Answer: We have re-analysed the data in detail and concentrated on citrate versus control groups and show the statistics for all the remaining groups (Fig. 2A; Supp. Fig. 6&7). Thanks to that we have determined that incubation of prostate cancer cells with citrate reduces intracellular levels of the metabolites known to be used for intracellular citrate synthesis (such as glutamine, glutamate, serine or glycine) which is in line with our hypothesis that part of the extracellular citrate is supplied to cancer cells from outside, p9. We are grateful for this comment.

I remain confused by Figure 2A, the level of consistency between the rows is low. The statistical analysis is not comprehensible and not broken down for each metabolite. The separation of the groups in the PCA plot is SF6 is not clear and the values for citrate + gluconate are very spread, which does not inspire confidence.

4. Figure 3A&D do not show convincing differences in EMT regulators. The authors should provide

quantification for multiple (three or more) biological replicates. Furthermore, the biological significance of such small changes is hard to gauge.

Answer: We have now performed the analysis as suggested by the Reviewer and included quantifications of the biological replicates not only for this part of the study but also for all the remaining parts where western blot analysis was used (Fig. 3A&D, Supp. Fig. 11). We do see consistent changes of the pattern of metabolic characteristics and EMT marker expression. Although, we do agree that not all the markers show significant changes but together with the metabolomics/metabolic and morphological data they stay in line with the observation that incubation with extracellular citrate stimulates acquisition of a more aggressive character of the cells as compared to the control.

OK

5. The differences in cell morphology in the SEM images in Figure 3B are not clear. The authors should provide some morphological quantification to back up their claims.

Answer: We have now analysed the pictures according to the papers dealing with the issue of invasion and colonisation and the data are presented in Fig. 3B and also in Supp. Fig. 10.

OK, but which papers and which metrics were used to classify single or ameboid cells from the EM images.

6. If performed properly, then intrasplenic injection of cancer cells followed by splenectomy should not result in the growth of tumors at the wound site. The fact that 60% of control mice develop tumors at their wounds is concerning.

Answer: L3.6pl cells are a very aggressive cancer cell type and some wound tumours are known to occur. We now discuss this result and show the literature supporting our statement, p. 13.

I remain concerned about this. Certainly, our local vet would not tolerate this. I would suggest removing panel B.

7. Gluconate is a pleiotropic molecule and cannot be considered a specific perturbation. I realize that it might not be totally straightforward, but the authors need to devise and use a more specific way of targeting pmCiC. Can Crispr or RNAi be used against the exon that results in plasma membrane targeting? Thereby leaving the mitochondrial isoform unaffected.

Answer: This is of course a very important question and for several years we have struggled to achieve specific silencing of the pmCiC. We have tried the following: 1. siRNA. We had siRNA designed by Eurofins and used all the options they suggested, however, due to a relatively small sequence difference between the two proteins and exceptionally high content of Cs and Gs none of the possibilities offered were optimal. Nonetheless, we used transient and permanent transfections, unfortunately both proteins pmCiC and mCiC expression has been affected. For this reason, we have used transient siRNA transfection for the citrate transport studies only as in this case decrease of mCiC would have no influence on the results (Mazurek et al., 2010; Mycielska et al., 2018). 2. We hoped that using transient transfections and adjusting the amount of the siRNA we could control more the effects on pmCiC versus mCiC. We performed additional studies to choose the most promising concentrations of the siRNA and performed animal studies using AteloGene (Koken) kit for transient transfections. Unfortunately, the reduction in mCiC was still prominent, which would make the obtained results unspecific. 3. CRISPR. We have chosen the first exon of the pmCiC to check several possibilities of CRISPR-Cas9. We have screened many clones and in all cases, reduction in pmCiC was accompanied by reduced mCiC expression. Importantly we have not found any clones with no expression of pmCiC suggesting that complete removal is lethal

to the cells consistent with an interdependent regulation of these two genes. To conclude, we are still not sure how pmCiC/mCiC expression is specifically regulated. Our results suggest that at the first steps, the two genes might be transcribed together and additional regulation is employed which allows for selecting one or the other for further processing. Unfortunately, this is a complex issue and out of the scope of the present study. Gluconate specificity Answer: Gluconate is plasma membrane impermeable, has been accepted as a physiologically neutral substance (by FDA), and is used widely in biology and medicine. In medicine gluconate is used as a carrier of metals such as Zn^{2+} , Mg^{2+} or Ca^{2+} even at very high concentrations (infusions of Ca^{2+} gluconate contain 1 g of gluconate). In electrophysiology and transport experiments gluconate is routinely used to replace chloride because of its lack of effects on the cells. Moreover, our current experiments depicted in the manuscript determined that gluconate affects citrate transport in cancer cells only, even though pmCiC expression occurs also in prostate epithelial cells and CAFs but with a 4 function of citrate export. Indeed, when applied alone (without citrate) at very high and totally unphysiological concentrations gluconate seems to have an effect on cancer cell mitochondria possibly through forced entrance through pmCiC (gluconate has a similar structure to citrate, therefore at high concentrations and in the absence of citrate some of it might be transported through pmCiC). In the present ms we show also that pmCiC has two binding sites for gluconate which are not the same as the citrate binding site (Fig. 2C). One of them is located in its N-terminus which is hanging freely, most likely in the extracellular space. This N-terminus is long and unusually heavily charged suggesting that it has a role either in regulating citrate transport, trafficking or also potentially some signalling/receptor functions. Binding of gluconate will affect the structure of N-terminus which might be followed by some additional effects. However, it still means that the only affected protein by gluconate is pmCiC and only in cancer cells, therefore, making gluconate a rather specific inhibitor. We are in the process of studying pmCiC structure and its functions further as well as preparing for a drug screen in cancer cells and CAFs, however, this issue is outside the scope of this manuscript.

OK, I accept this.

8. Why discuss the ATAC seq data if it is inconclusive and peripheral to the main message?

Answer: These data have now been removed.

Good

9. The data in Figure 6 are anecdotal and not really convincing.

Answer: Fig. 6 includes 6 examples of pmCiC immunohistochemical staining of pancreatic and gastric cancers. These cancers are known to have strong desmoplastic fibrosis with abundant cancer associated fibroblasts (CAFs) and accordingly we see the pmCiC positivity in numerous CAFs in these tissues. For the purpose of this study we analysed 25 gastric carcinomas (diffuse and intestinal type) as well as 25 pancreatic ductal adenocarcinomas. As shown in figure 6 we could see the most prominent pmCiC expression in cancer cells at the invasion front but also in the spindle shaped cells of the stromal reaction, which are mainly CAFs. In addition, we have stained 32 samples of head and neck squamous cell cancer (HSNCC), 59 invasive/non-invasive bladder cancer, 156 metastatic and non-metastatic breast cancer (hormone receptor positive/negative and HER-2 types), 39 bone and lymph node metastasis of breast and prostate cancer, 6 Glioblastomas, 56 metastatic and non - metastatic colon cancer tissues and found an expression of pmCiC in cancer cells and also in cells of the tumor microenvironment (TME). We are preparing an in depth analysis of the tissues regarding pmCiC expression in cancer cells and in the TME for a separate manuscript as we believe this is outside the scope of the present project. We have now added higher magnifications of the immunohistochemical staining to Fig. 6 to better highlight the expression of pmCiC in the TME. Although we agree with the reviewer that more staining is needed to better

understand the role of pmCiC in CAF 's in human cancer, we are convinced that these data together with the in vitro and animal experiments show that pmCiC is present in CAF's and plays a role in the complex interplay of cancer cells with the TME.

This information is re-assuring, but the authors need to provide some blinded quantification that summarizes the analysis of the large number of samples that they say they have looked.

Reviewer #2 (Comments to the Authors (Required)):

The authors have added some new results in response to reviewers' comments, which somewhat improves the manuscript. The manuscript is still not easy to follow, partially due to the complex treatments and very particular experimental settings to be tested. It probably can be improved by a different and more clear writing style. Nevertheless, the manuscript does provide original and meaningful data, and thus I would support it if other reviewers and the editor are favorable towards acceptance for publication.

Reviewer #3 (Comments to the Authors (Required)):

The authors have satisfactorily addressed most of my questions.

Dear Dr. Bhatt,

Thank you very much for your decision letter which we have received on the 2.03.21 regarding the revision of our manuscript entitled "Cancer-associated cells release citrate to support tumour metastatic progression". We are grateful for additional comments from the Reviewers. Below you will find our detailed answers with appropriate amendments to the manuscript. We have followed the formatting guidelines and corrected the manuscript according to the points raised in your letter. We have included all the figures containing tables as TIFs and also in editable forms.

We hope you will find the changes satisfactory and look forward to hearing from you,

Kind regards,

Maria Mycielska and Sebastian Haferkamp

Reviewer #1 (Comments to the Authors (Required)):

Overall, this work is improved and there is enough in it to warrant publication. However, it remains let down by confusing presentation and there are numerous fairly simple things that should be done to improve clarity and rigor. I realize that this is a bit long winded, but I have included both my original comment, the response, and my new response below.

1. Figure 1B top panel does not show a convincing difference in pmCiC + or - citrate. Answer: We now show the absolute values of citrate released by fibroblasts into the media - Fig. 1B, p. 6-7.

OK

2. It would be helpful to know the relative levels of cytokines produced by the cancer cells and fibroblasts. It is not clear if the pixel density measurement allows for comparison between the two plots. Answer: The assay was used according to the manufacturer instructions and analysed using supplied positive and negative controls for all the assays. We used the same control media for all conditions tested, therefore we believe it is possible to use the values shown in the figures for the comparative analysis of the relative values of cytokines in cancer cells and fibroblasts, p. 21.

Question: *The authors seem to have only partially answered the question. It is also critical that the exposure times are equivalent and the analyses performed in parallel. Having said that, the analysis presented is OK so long as the authors only make claims for each cell type alone +/- citrate and do not try to claim absolute differences in the levels between cancer cells and CAFs.*

Answer: To clarify the issue, all the samples were measured in parallel with all the membranes prepared at the same time. Pictures of all the membranes were taken at several time points. The clearest pictures were taken for analysis. However, every membrane contains a control to which the results are normalised, therefore these differences are accounted for. Nonetheless, the statistics were performed on the results from the same cell type +/-citrate, we never compared absolute differences between the different cell types, we only included graphs showing a general trend for particular cell types and conditions. This is now explained in the M&M section page 21.

3. Figure 2A shows no consistent difference between the + and - citrate samples. Also, the legend is not sufficient to fully understand what is being shown. Is each row a replicate? Answer: We have re-analysed the data in detail and concentrated on citrate versus control groups and show the statistics for all the remaining groups (Fig. 2A; Supp. Fig. 6&7). Thanks to that we have determined that

incubation of prostate cancer cells with citrate reduces intracellular levels of the metabolites known to be used for intracellular citrate synthesis (such as glutamine, glutamate, serine or glycine) which is in line with our hypothesis that part of the extracellular citrate is supplied to cancer cells from outside, p9. We are grateful for this comment.

Question: *I remain confused by Figure 2A, the level of consistency between the rows is low. The statistical analysis is not comprehensible and not broken down for each metabolite. The separation of the groups in the PCA plot is SF6 is not clear and the values for citrate + gluconate are very spread, which does not inspire confidence.*

Answer: We performed all analyses of metabolites for the sample set obtained after cell line cultivation at four conditions (four group data set) according to state-of-the-art procedures for metabolomics. Because of large number of independent metabolites analysed we refrained from two-way ANOVA as suggested by the reviewer as this would not provide unbiased differentiation and could rather result in overfitting or high FDR. Therefore, we first presented rooted heatmap for two data groups (Figure 2A). Further, we used multivariate approach with partial least squares discriminant analysis (PLS-DA) to estimate the group separation (Supplementary Figure 7 - SF7, it is not the Supplementary Figure 6 as stated by the reviewer). As stated in the legend of this SF7 the quality parameters were supporting the group separation as the R2Y (cumulative) was calculated to be 0.85 and Q2Y was calculated to be 0.199, whereas the RMSEE is estimated to be 0.189. This low Q2Y value indicates small differences between the groups. We would like to stress that these data were cross-validated seven times with 2000-fold permutations. Consequent analyses of variables of importance in projection (VIP) listed 80 metabolites which passed the thresholds confirming metabolite significance from Fig. 2A. All this information is included in the M&M section, pp 28-29.

4. Figure 3A&D do not show convincing differences in EMT regulators. The authors should provide quantification for multiple (three or more) biological replicates. Furthermore, the biological significance of such small changes is hard to gauge. **Answer:** We have now performed the analysis as suggested by the Reviewer and included quantifications of the biological replicates not only for this part of the study but also for all the remaining parts where western blot analysis was used (Fig. 3A&D, Supp. Fig. 11). We do see consistent changes of the pattern of metabolic characteristics and EMT marker expression. Although, we do agree that not all the markers show significant changes but together with the metabolomics/metabolic and morphological data they stay in line with the observation that incubation with extracellular citrate stimulates acquisition of a more aggressive character of the cells as compared to the control.

OK

5. The differences in cell morphology in the SEM images in Figure 3B are not clear. The authors should provide some morphological quantification to back up their claims. **Answer:** We have now analysed the pictures according to the papers dealing with the issue of invasion and colonisation and the data are presented in Fig. 3B and also in Supp. Fig. 10.

Question: *OK, but which papers and which metrics were used to classify single or amoeboid cells from the EM images.*

Answer: We have performed our analysis based on the papers by Morley et al., 2014 (Trading in your spindles for blebs: the amoeboid tumor cell phenotype in prostate cancer) as well as Friedl and Wolf 2003 (Tumour-cell invasion and migration: diversity and escape mechanisms; the references are included in the manuscript text, p. 11). Cells with round body and a distinct single pseudopodium were considered as amoeboidal (as in the Fig. S10), cells with no contact with other cells or cells with

less than 20% contact with others were considered as single. This description is now included into M&M section, page 27.

6. If performed properly, then intrasplenic injection of cancer cells followed by splenectomy should not result in the growth of tumors at the wound site. The fact that 60% of control mice develop tumors at their wounds is concerning. Answer: L3.6pl cells are a very aggressive cancer cell type and some wound tumours are known to occur. We now discuss this result and show the literature supporting our statement, p. 13.

Question: *I remain concerned about this. Certainly, our local vet would not tolerate this. I would suggest removing panel B.*

Answer: Panel B has now been removed.

7. Gluconate is a pleiotropic molecule and cannot be considered a specific perturbation. I realize that it might not be totally straightforward, but the authors need to devise and use a more specific way of targeting pmCiC. Can Crispr or RNAi be used against the exon that results in plasma membrane targeting? Thereby leaving the mitochondrial isoform unaffected. Answer: This is of course a very important question and for several years we have struggled to achieve specific silencing of the pmCiC. We have tried the following: 1. siRNA. We had siRNA designed by Eurofins and used all the options they suggested, however, due to a relatively small sequence difference between the two proteins and exceptionally high content of Cs and Gs none of the possibilities offered were optimal. Nonetheless, we used transient and permanent transfections, unfortunately both proteins pmCiC and mCiC expression has been affected. For this reason, we have used transient siRNA transfection for the citrate transport studies only as in this case decrease of mCiC would have no influence on the results (Mazurek et al., 2010; Mycielska et al., 2018). 2. We hoped that using transient transfections and adjusting the amount of the siRNA we could control more the effects on pmCiC versus mCiC. We performed additional studies to choose the most promising concentrations of the siRNA and performed animal studies using AteloGene (Koken) kit for transient transfections. Unfortunately, the reduction in mCiC was still prominent, which would make the obtained results unspecific. 3. CRISPR. We have chosen the first exon of the pmCiC to check several possibilities of CRISPR-Cas9. We have screened many clones and in all cases, reduction in pmCiC was accompanied by reduced mCiC expression. Importantly we have not found any clones with no expression of pmCiC suggesting that complete removal is lethal to the cells consistent with an interdependent regulation of these two genes. To conclude, we are still not sure how pmCiC/mCiC expression is specifically regulated. Our results suggest that at the first steps, the two genes might be transcribed together and additional regulation is employed which allows for selecting one or the other for further processing. Unfortunately, this is a complex issue and out of the scope of the present study. Gluconate specificity Answer: Gluconate is plasma membrane impermeable, has been accepted as a physiologically neutral substance (by FDA), and is used widely in biology and medicine. In medicine gluconate is used as a carrier of metals such as Zn²⁺, Mg²⁺ or Ca²⁺ even at very high concentrations (infusions of Ca²⁺ gluconate contain 1 g of gluconate). In electrophysiology and transport experiments gluconate is routinely used to replace chloride because of its lack of effects on the cells. Moreover, our current experiments depicted in the manuscript determined that gluconate affects citrate transport in cancer cells only, even though pmCiC expression occurs also in prostate epithelial cells and CAFs but with a 4 function of citrate export. Indeed, when applied alone (without citrate) at very high and totally unphysiological concentrations gluconate seems to have an effect on cancer cell mitochondria possibly through forced entrance through pmCiC (gluconate has a similar structure to citrate, therefore at high concentrations and in the absence of citrate some of it might be transported through pmCiC). In the present ms we show also that pmCiC has two binding sites for gluconate

which are not the same as the citrate binding site (Fig. 2C). One of them is located in its N-terminus which is hanging freely, most likely in the extracellular space. This N-terminus is long and unusually heavily charged suggesting that it has a role either in regulating citrate transport, trafficking or also potentially some signalling/receptor functions. Binding of gluconate will affect the structure of N-terminus which might be followed by some additional effects. However, it still means that the only affected protein by gluconate is pmCiC and only in cancer cells, therefore, making gluconate a rather specific inhibitor. We are in the process of studying pmCiC structure and its functions further as well as preparing for a drug screen in cancer cells and CAFs, however, this issue is outside the scope of this manuscript.

OK, I accept this.

8. Why discuss the ATAC seq data if it is inconclusive and peripheral to the main message? Answer: These data have now been removed.

Good

9. The data in Figure 6 are anecdotal and not really convincing. Answer: Fig. 6 includes 6 examples of pmCiC immunohistochemical staining of pancreatic and gastric cancers. These cancers are known to have strong desmoplastic fibrosis with abundant cancer associated fibroblasts (CAFs) and accordingly we see the pmCiC positivity in numerous CAFs in these tissues. For the purpose of this study we analysed 25 gastric carcinomas (diffuse and intestinal type) as well as 25 pancreatic ductal adenocarcinomas. As shown in figure 6 we could see the most prominent pmCiC expression in cancer cells at the invasion front but also in the spindle shaped cells of the stromal reaction, which are mainly CAFs. In addition, we have stained 32 samples of head and neck squamous cell cancer (HSNCC), 59 invasive/non-invasive bladder cancer, 156 metastatic and non-metastatic breast cancer (hormone receptor positive/negative and HER-2 types), 39 bone and lymph node metastasis of breast and prostate cancer, 6 Glioblastomas, 56 metastatic and non - metastatic colon cancer tissues and found an expression of pmCiC in cancer cells and also in cells of the tumor microenvironment (TME). We are preparing an in depth analysis of the tissues regarding pmCiC expression in cancer cells and in the TME for a separate manuscript as we believe this is outside the scope of the present project. We have now added higher magnifications of the immunohistochemical staining to Fig. 6 to better highlight the expression of pmCiC in the TME. Although we agree with the reviewer that more staining is needed to better understand the role of pmCiC in CAF 's in human cancer, we are convinced that these data together with the in vitro and animal experiments show that pmCiC is present in CAF's and plays a role in the complex interplay of cancer cells with the TME.

Question: *This information is re-assuring, but the authors need to provide some blinded quantification that summarizes the analysis of the large number of samples that they say they have looked.*

Answer: To address this criticism we have performed additional assessment of the breast, gastric and pancreatic human tissues and the results are presented in Fig. 6G, p. 15. We believe that indeed, this analysis makes the conclusions of our study stronger and we are grateful for this comment.

Reviewer #2 (Comments to the Authors (Required)):

The authors have added some new results in response to reviewers' comments, which somewhat improves the manuscript. The manuscript is still not easy to follow, partially due to the complex treatments and very particular experimental settings to be tested. It probably can be improved by a different and more clear writing style. Nevertheless, the manuscript does provide original and

meaningful data, and thus I would support it if other reviewers and the editor are favorable towards acceptance for publication.

Answer: We do agree with the Reviewer that due to the complexity of this study it is difficult to make the manuscript easy to follow. We have discussed it with all the co-authors representing different expertise and tried to choose the best way to present our findings and experimental procedures. We have gone through the whole manuscript again and tried to make some small final adjustments that could facilitate understanding of our work. We hope that our study is now easier to understand.

Reviewer #3 (Comments to the Authors (Required)):

The authors have satisfactorily addressed most of my questions.

March 10, 2021

RE: Life Science Alliance Manuscript #LSA-2020-00903-TRR

Dr. Maria Mycielska
University Hospital Regensburg
Surgery
Franz-Josef-Strauß-Allee 11
Regensburg 93053
Germany

Dear Dr. Mycielska,

Thank you for submitting your Research Article entitled "Cancer-associated cells release citrate to support tumour metastatic progression". It is a pleasure to let you know that your manuscript is now accepted for publication in Life Science Alliance. Congratulations on this interesting work.

DISTRIBUTION OF MATERIALS:

Again, congratulations on a very nice paper. I hope you found the review process to be constructive and are pleased with how the manuscript was handled editorially. We look forward to future exciting submissions from your lab.

Sincerely,

Shachi Bhatt, Ph.D.

Executive Editor

Life Science Alliance

<https://www.lsjournal.org/>

Interested in an editorial career? EMBO Solutions is hiring a Scientific Editor to join the international Life Science Alliance team. Find out more here -

https://www.embo.org/documents/jobs/Vacancy_Notice_Scientific_editor_LSA.pdf